# Evaluation of a New Merged Sea-Ice Concentration Dataset at 1 km Resolution from Thermal Infrared and Passive Microwave Satellite Data in the Arctic

**Valentin Ludwig** [1,*], **Gunnar Spreen** [1] **and Leif Toudal Pedersen** [2]

[1]   Institute of Environmental Physics, University of Bremen, Otto-Hahn-Allee 1, 28359 Bremen, Germany; gunnar.spreen@uni-bremen.de

[2]   DTU-Space, Technical University of Denmark, DK-2800 Lyngby, Denmark; ltp@space.dtu.dk

*   Correspondence: vludwig@uni-bremen.de

**Abstract:** Sea-ice concentration (SIC) data with fine spatial resolution and spatially continuous coverage are needed, for example, for estimating heat fluxes. Passive microwave measurements of the Advanced Scanning Microwave Radiometer 2 (AMSR2) offer spatial continuity, but are limited to spatial resolutions of 5 km and coarser. Thermal infrared data of the Moderate Resolution Imaging Spectroradiometer (MODIS) provide a spatial resolution of 1 km, but are limited to cloud-free scenes. We exploit the benefits of both and present a merged SIC dataset with 1 km spatial resolution and spatially continuous coverage for the Arctic. MODIS and AMSR2 SIC are retrieved separately and then merged by tuning the MODIS SIC to preserve the mean AMSR2 SIC. We first evaluate the variability of the dynamically retrieved MODIS ice tie-point. Varying the starting position of the area used for the tie-point retrieval changes the MODIS SIC by on average 1.9%, which we mitigate by considering different starting positions and using the average as ice tie-point. Furthermore, the SIC datasets are evaluated against a reference dataset derived from Sentinel-2A/B reflectances between February and May 2019. We find that the merged SIC are 1.9% smaller than the reference SIC if thin ice is considered as ice and 4.9% higher if thin ice is considered as water. There is only a slight bias (0.3%) between the MODIS and the merged SIC; however, the root mean square deviation of 5% indicates that the two datasets do yield different results. In an example of poor-quality MODIS SIC, we identify an unscreened cloud and high ice-surface temperature as reasons for the poor quality. Still, the merged SIC are of similar quality as the passive microwave SIC in this example. The benefit of merging MODIS and AMSR2 data is demonstrated by showing that the finer resolution of the merged SIC compared to the AMSR2 SIC allows an enhanced potential for the retrieval of leads. At the same time, the data are available regardless of clouds. Last, we provide uncertainty estimates. The MODIS and merged SIC uncertainty are between 5% and 10% from February to April and increase up to 25% (merged SIC) and 35% (MODIS SIC) in May. They are identified as conservative uncertainty estimates.

**Keywords:** arctic; sea-ice concentration; fine spatial resolution; merging; thermal infrared; passive microwave; uncertainties; MODIS; AMSR2

## 1. Introduction

The Arctic region is of high importance for the Earth's climate. Its influence extends far outside the polar regions and affects the weather in the mid-latitudes as well as the global radiative budget. Sea-ice concentration (SIC), the fraction of a given ocean area covered by sea ice, has been monitored from space since 1972 with the Electrically Scanning Microwave Radiometer (ESMR) and routinely since 1979 with two-daily Arctic-wide coverage until 1987 and daily Arctic-wide coverage from 1987

onwards. The spatial resolutions of the first passive microwave radiometers were about 50 km of the 19 GHz and 25 km of the 37 GHz channels of the Scanning Multichannel Microwave Radiometer (SMMR). Nowadays, spatial resolutions of the 89 GHz channels of the Advanced Microwave Scanning Radiometer 2 (AMSR2), which are used in this study, are less than 5 km. The resolutions are given here as the instantaneous field of view.

While these spatial resolutions are fine enough for climate monitoring, finer resolutions are desirable for model input, navigation and studies at regional scale. SIC data with a spatial resolution of 1 km and finer can be retrieved from thermal infrared measurements. These are currently available operationally from the Moderate Resolution Imaging Spectroradiometer (MODIS) instruments aboard the Terra and Aqua satellites, from the Visible Infrared Imaging Radiometer Suite (VIIRS) aboard Suomi National Polar-orbiting Partnership (NPP) and from the Sea and Land Surface Temperature Radiometer (SLSTR) and the Ocean Land Color Instrument (OLCI) aboard the Sentinel-3A/B satellites. The data are available since 2000 (Terra), 2002 (Aqua), 2011 (VIIRS) and 2016/2018 (Sentinel-3A/B). MODIS Aqua data are used in this study.

Thermal infrared SIC algorithms use the temperature contrast between the ice and water surface. The quantity which is inferred from thermal infrared measurements is called "potential open water". Potential open water is the open-water fraction of a pixel which would be required for the pixel to have the ice-surface temperature (IST) measured by the satellite, assuming that the pixel comprises water and ice which is thick enough to prevent oceanic heat flux [1,2]. The inverse of the potential open water is used as thermal infrared SIC in this study. The temperature contrast can also be used for lead retrieval, as is done in [3]. Under the assumption that leads (linear openings in the ice) represent the high end of the surface temperature anomaly histogram, they evaluate different segmentation approaches to get binarised lead/no lead images from which daily lead composite maps are derived.

While thermal infrared measurements (MODIS measurements hereafter) have a finer spatial resolution than passive microwave measurements (AMSR2 measurements hereafter), they are limited to cloud-free scenes. Furthermore, they are more sensitive to sea-ice thickness than AMSR2 measurements, which hinders the development of a spatially continuous SIC dataset with the correct magnitude from MODIS measurements alone. Generally speaking, MODIS SIC capture the spatial variability of SIC well, but do often not reflect the correct magnitude. The correct magnitude, together with a spatially continuous coverage, can be obtained from AMSR2 radiometers. This comes at the cost of a coarser spatial resolution. We use the best of the two measurements by merging them and obtaining a spatially continuous dataset at 1 km by 1 km resolution which combines the small-scale features resolved by the MODIS data with the correct magnitude retrieved from the AMSR2 measurements. If no MODIS data are available, the actual resolution decreases to 5 km by 5 km, which corresponds to one grid cell of AMSR2 measurements at the 89 GHz channels.

One advantage of our merged SIC dataset over AMSR2 SIC is the enhanced potential for lead retrieval, where leads show up as reduced SIC. The enhanced potential owes to the fact that AMSR2 SIC show 100% SIC for ice thicker than 10 cm [4,5], while MODIS data show leads also if they are covered by thicker ice. Leads are of large importance for heat flux [3,6] and ocean–air gas exchange [7]. Furthermore, the they are important for navigation. For use as model input, spatial continuity is required, which our dataset has due to the inclusion of AMSR2 SIC. We thus provide a dataset which is better suited for calculating ocean–air fluxes than AMSR2 SIC alone and better suited for model input than MODIS SIC alone.

We have been producing the merged SIC dataset operationally between October 2019 and May 2020. During this time, more than 60% of the pixels over sea ice were cloud-free at least once per day on 84% of the days. More than 80% of the pixels were cloud-free on 61% of the days. The actual coverage may have been even better, but sometimes not all MODIS data were available yet at the time of processing. This shows that the operational provision of an Arctic-wide 1 km SIC dataset is possible, which motivates the intercomparison and uncertainty assessment in this study.

First results of the merged dataset are presented in [8], together with a case study of a polynya which opened north of Greenland in February 2018. In this study, the authors of [8] show that, especially during the opening of the polynya, the merged SIC dataset is better suited for monitoring the situation than either of the input datasets alone. While the work in [8] focuses on the application of the merged dataset, this study presents an extensive analysis of the sensitivity of the MODIS SIC dataset to the choice of the MODIS ice tie-point, an intercomparison with an independently derived SIC dataset from Sentinel-2 reflectances and an uncertainty estimate. Specifically, the following questions shall be answered.

1. How sensitive is the merged sea-ice concentration towards the choice of the MODIS ice tie-point?
2. How do the single-sensor and merged SIC datasets compare with each other and the independently derived Sentinel-2 SIC dataset?
3. What are the uncertainties of the merged, MODIS and AMSR2 datasets?

Section 2 will present the data and methods used in the study. Sections 3.1–3.3 will present the analysis of the sensitivity towards the choice of the MODIS ice tie-point, the intercomparison study and the uncertainty estimate, respectively. Section 4 discusses the findings and puts them into perspective to other studies. Section 5 will summarise the content, and Section 6 will present the answers to the research questions posed above. Finally, Section 7 will give an outlook for future work.

## 2. Material and Methods

### 2.1. MODIS Data

NASA's MODIS instrument, (https://modis.gsfc.nasa.gov/, last access 12 September 2020) is borne by two satellites, namely, Terra (since 2000) and Aqua (since 2002). As Aqua flies in the same satellite constellation as AMSR2's platform Global Change Observation Mission–Water (GCOM-W1) with a 4-minute time lag, we exclusively use MODIS Aqua data. MODIS operates in the visible and thermal infrared spectrum. It provides measurements in 36 wavelength bands centred at wavelengths between 645 nm and 14.235 µm. The MYD29 IST dataset which we use has been developed at Goddard Space Flight Center [9] and is provided by the National Snow and Ice Data Center (NSIDC) at https://n5eil01u.ecs.nsidc.org/MOSA/MYD29.006/ (last access 12 September 2020). The brightness temperatures of band 31 (centred at 11.03 µm) and band 32 (centred at 12.02 µm) are used to derive the IST by applying a split-window technique [10]. The authors of [10] apply a conservative cloud mask by masking only those pixels which the MODIS cloud mask (MYD35_L2, [11] https://ladsweb.modaps.eosdis.nasa.gov/archive/allData/61/MYD35_L2/, last access 12 September 2020) classifies as "confident cloudy". We apply a stricter cloud mask by also discarding pixels which are labelled as "probably cloudy" and "probably clear", thus only tolerating pixels which are labelled "confident clear". Additionally, pixels over land, inland water and the open ocean are masked out by both cloud masks. The MYD29 data are sliced into granules which cover ~5 min. Each granule has a spatial dimension of 2030 km by 1354 km. The grid spacing is 1 km. Further details on the MYD29 dataset are given in [10], at https://nsidc.org/data/MOD29/versions/6 (last access 12 September 2020) and in the Algorithm Theoretical Baseline Document at https://modis.gsfc.nasa.gov/data/atbd/atbd_mod10.pdf (last access 12 September 2020). For each Sentinel-2 scene, we downloaded all MODIS granules which intersected with the scene on the same day (997 granules in total). For the assessment of the MODIS ice tie-point (Section 3.1) and the uncertainty estimates (Section 3.3), all of these MODIS granules are used. For the intercomparison with the Sentinel-2 SIC (Section 3.2), only the MODIS granules pertaining to the swath with the smallest time lag towards the Sentinel-2 scene are used. If the time lag exceeds 120 min, no intercomparison is done. For the merging, the granules are gridded to a north polar stereographic grid with the latitude of true scale at 70°N, also known and hereafter referred to as the NSIDC grid (https://nsidc.org/data/polar-stereo/ps_grids.html, last access 12 September 2020). For the intercomparison with the Sentinel-2 SIC, they are resampled to a Transverse Mercator projection for consistency with the Sentinel-2 data.

## 2.2. ASI-AMSR2 Sea-Ice Concentration

The Advanced Microwave Scanning Radiometer 2 (AMSR2) aboard the Japan Aerospace Exploration Agency's (JAXA's) GCOM-W1 measures horizontally and vertically polarised electromagnetic radiation in seven microwave frequency bands between 6.9 GHz and 89 GHz. The polarisation difference at 89 GHz is used by the Arctic Radiation and Turbulence Interaction Study (ARTIST) Sea Ice algorithm (ASI). The ASI-AMSR2 SIC is provided as a daily gridded dataset at 6.25 km and 3.125 km grid spacing at https://seaice.uni-bremen.de (last access 12 September 2020). For a better temporal consistency with the MODIS data, we use swath data processed internally. For the merged dataset, they are interpolated to the same projection and grid spacing as the MODIS data. For the intercomparison with the Sentinel-2 SIC, they are resampled to a Transverse Mercator projection for consistency with the Sentinel-2 data.

## 2.3. Sentinel-2 Reflectances

Sentinel-2 reflectances are used to create a reference SIC dataset for evaluation of the merged, MODIS and AMSR2 SIC. The EU Copernicus Sentinel-2A/B satellites were launched in June 2015 (Sentinel-2A) and March 2017 (Sentinel-2B). They follow a sun-synchronous orbit at 786 km altitude and 10:30 am equator crossing time. Their phase is shifted by 180°. Their payload, the Multispectral Instrument (MSI), features 13 bands in the visible and shortwave infrared spectral region. To obtain a reference dataset, we downloaded 79 cloud-free scenes from https://scihub.copernicus.eu/ (last access 12 September 2020). We made sure that they are cloud-free by checking every scene visually. Figure 1 and Table 1 offer a spatial and temporal overview over the scenes. Scenes with more than two hours time lag towards the next MODIS/AMSR2 overflight are not used for the intercomparison, but are included in the paper because of their possible relevance for other validation studies. The scenes were recorded in the East Siberian, Laptev, Kara, Barents, Beaufort Sea and in the Fram Strait between 22 February 2019 and 27th May 2019. The scenes comprise mainly first-year ice with smaller amounts of young ice. Ice-type maps based on Advanced Scatterometer (ASCAT) and AMSR2 data ([12–14], https://seaice.uni-bremen.de/multiyear-ice-concentration/, last access 12 September 2020) show some multiyear ice in the East Siberian, Fram Strait and Beaufort Sea (not explicitly shown here). For this study, Level 1 C reflectances at 665 nm (band 4) with a spatial resolution of 10 m are used. Level 1 C data are top-of-the-atmosphere reflectances which are projected onto a Universal Transverse Mercator projection.

**Table 1.** Number of Sentinel-2 scenes for the respective region and month. If there is a number in brackets, it gives the number of scenes after discarding Sentinel-2 scenes for which no cloud-free MODIS measurement is available within two hours of the acquisition. In those without brackets all scenes could be used.

| Region | Feb | Mar | Apr | May | Total |
|---|---|---|---|---|---|
| East Siberian Sea | 0 | 10 (3) | 8 (5) | 0 | 18 (8) |
| Laptev Sea | 3 | 4 | 2 | 10 | 19 |
| Kara Sea | 3 | 7 | 10 | 9 (8) | 29 (28) |
| Barents Sea | 0 | 1 | 0 | 2 | 3 |
| Fram Strait | 0 | 1 (0) | 2 (1) | 1 | 4 (2) |
| Beaufort Sea | 1 | 1 | 2 | 2 | 6 |
| All | 7 | 24 (16) | 24 (20) | 24 (23) | 79 (66) |

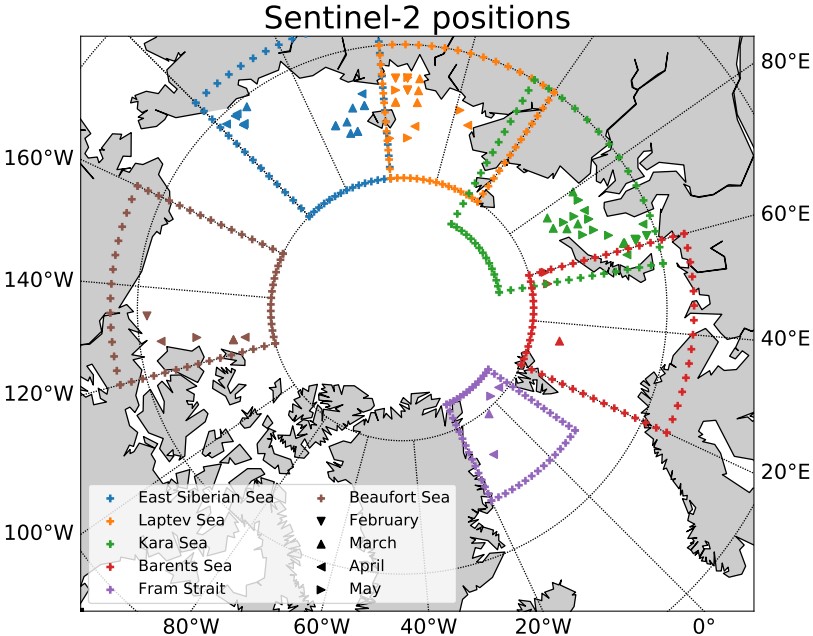

**Figure 1.** Locations of the Sentinel-2 scenes. The polygons mark the regions which are used for defining the marginal seas. Regions are indicated by colours and Months are indicated by the orientation of the triangles (see legend for details). All in all, 79 scenes are used.

### 2.4. MODIS Sea-Ice Concentration Retrieval

For the MODIS SIC retrieval, accurate knowledge of the background temperatures is crucial. The background temperatures, hereafter referred to as tie-points, are the reference temperatures which a pixel would have if it were covered completely by open water (water tie-point $tp_{water}$) or covered completely by sea ice (ice tie-point $tp_{ice}$). For the tie-point retrieval, we follow the approach of the works in [1,8].

The choice of $tp_{water}$ is straightforward: we assume that the open water is constantly at the freezing point of salt water, $-1.8\,°C$. If it would be colder, the water would freeze to sea ice until thermal equilibrium is reached. If it would be warmer, it would melt the surrounding the sea ice and thereby cool down to the freezing point. However, until equilibrium is reached other water temperatures can also be observed, which is part of the $tp_{water}$ uncertainty.

The choice of $tp_{ice}$ is less straightforward. The IST follows the air temperature and thus has large spatial and temporal variability, which renders it impossible to choose an Arctic-wide fixed $tp_{ice}$. To account for the local variability, each pixel is assigned its own $tp_{ice}$. For this, the IST field is split into cells of 48 pixels by 48 pixels. We assume only linear IST variation within these cells. Each cell is split into 3 by 3 subcells of 16 pixels by 16 pixels. Within a subcell, we assume negligible IST variations. In the next step, the 25th percentile of the IST field in the subcell is selected as preliminary $tp_{ice}$, so that there are 9 preliminary $tp_{ice,tmp}$ values per cell. Once the $tp_{ice,tmp}$ are selected, a linear regression with two variables is performed within the cell to express the $tp_{ice}$ as function of the x/y position within the cell:

$$tp_{ice}(x,y) = ax + by + c, \qquad (1)$$

where $x$ and $y$ are the indices of the respective pixel within the cell, $tp_{ice}(x,y)$ is $tp_{ice}$ at this pixel, and $a$, $b$ and $c$ are the regression coefficients. Subcells are discarded entirely if more than 70% of the pixels are covered by clouds and cells are discarded entirely if more than four subcells are discarded entirely. The authors of [15] investigate the sensitivity of the retrieval towards the required fraction of cloud-free pixels within one subcell, the required number of valid subcells in one cell and the percentile for the estimation of $tp_{ice,tmp}$. They confirm that the initial settings are a reasonable choice. So far, the approach directly follows that in [1]. They shift the cells by 48 pixels once the calculation is done,

so that each pixel is covered once. As in [8], we shift the cell by 1 pixel before repeating the calculation, so that each pixel is covered 48 times. The mean of the 48 iterations is then selected as $tp_{ice}$. Our approach allows to use the standard deviation of the 48 iterations as estimate for the uncertainty of $tp_{ice}$. Once $tp_{ice}$ for each pixel is determined, the IST is converted to SIC by linear interpolation:

$$SIC = \begin{cases} 1 & \text{if } IST < tp_{ice} \\ \frac{IST - tp_{water}}{tp_{ice} - tp_{water}} & \text{if } tp_{ice} < IST < tp_{water} \\ 0 & \text{if } IST > tp_{water} \end{cases} \tag{2}$$

No SIC retrieval is performed in the current version of our retrieval if $tp_{ice}$ exceeds 270 K.

The SIC calculation is performed on granule level. Subsequently, each granule is gridded to the NSIDC projection with 1 km grid spacing. Then, granules which belong to the same orbit are merged together, so that there is one MODIS SIC array per orbit.

### 2.5. Merging MODIS and AMSR2 Sea-Ice Concentration

After combining the gridded MODIS granules to one overflight, we select the corresponding AMSR2 overflight. As both platforms, Aqua and GCOM-W1, are part of the A-Train, the time lag is on the order of minutes. Our central assumption for the merging is that the MODIS SIC capture the variability at 1 km resolution correctly, but tend to underestimate the SIC due to the influence of thin ice thickness. Furthermore, they are limited by the presence of clouds. The AMSR2 SIC, on the other hand, are less sensitive to thin ice than the MODIS SIC, and thus retrieve the right magnitude at a spatial scale of 5 km and are available independently of cloud coverage. We thus tune the mean of the MODIS SIC in an area of 5 km by 5 km to match that of the AMSR2 data. This is done in the following way. First, the AMSR2 data are resampled/interpolated to a grid cell size of 1 km. Then, the difference between the mean MODIS SIC and the mean AMSR2 SIC in an area of 5 km by 5 km (i.e., 5 by 5 1 km-by-1 km pixels on the interpolated grid, not the original AMSR2 grid) is calculated:

$$\Delta_{SIC,5\,km} = SIC_{MODIS,5\,km} - SIC_{AMSR2,5\,km}, \tag{3}$$

where $SIC_{MODIS,5\,km}$ and $SIC_{AMSR2,5\,km}$ are the mean MODIS and AMSR2 SIC in the respective 5 km by 5 km cell and $\Delta_{SIC,5\,km}$ is their difference. This difference is now added to each MODIS pixel in this box such that it matches the mean of the AMSR2 pixels. In the final step, the cloud gaps are filled by the respective AMSR2 pixels:

$$SIC_{merged_{x,y}} = \begin{cases} SIC_{MODIS,5\,km_{x,y}} + \Delta_{SIC,5\,km} & \text{if } SIC_{MODIS,5\,km_{x,y}} \text{ available} \\ SIC_{AMSR2,5\,km_{x,y}} & \text{if } SIC_{MODIS,5\,km_{x,y}} \text{ not available} \end{cases} \tag{4}$$

Here, $SIC_{merged_{x,y}}$ is the merged SIC at the pixel with the coordinates $x, y$; $SIC_{MODIS,5\,km_{x,y}}$ and $SIC_{AMSR2,5\,km_{x,y}}$ are the MODIS and AMSR2 SIC at this pixel, respectively, and $\Delta_{SIC,5km}$ is defined as in Equation (4). A similar approach has been used in [16]. Different to them, we shift the 5 km by 5 km box by 1 pixel before the calculation is repeated for each pixel in the box. The mean is selected as merged SIC value. This mitigates the inaccuracy of choosing arbitrary starting positions for the merging, analogously to the approach which we chose for the $tp_{ice}$ retrieval in Section 2.4.

If the AMSR2 SIC are 100% in the entire scene, we allow SIC above 100% in some grid cells in order to resolve leads with reduced MODIS SIC which the AMSR2 did not show because they are overfrozen. By doing this the requirement of the merging procedure to keep the AMSR2 SIC mean (i.e., 100%) is fulfilled. In the end, however, the merged SIC is capped at 100%, which means that the means of the merged and the AMSR2 SIC are not entirely consistent in such cases. We accept this as a compromise for the enhanced potential to resolve leads. The above 100% SIC values before the truncation will be provided as additional information in our dataset.

### 2.6. Deriving Uncertainty Estimates

#### 2.6.1. MODIS Sea-Ice Concentration Uncertainty

Several uncertainty-afflicted parameters enter the MODIS SIC retrieval. The MODIS SIC uncertainty, $\sigma_{SIC_{MODIS}}$, is given by Gaussian error propagation applied to Equation (2):

$$\sigma_{SIC_{MODIS}} = \sqrt{Unc_{IST} + Unc_{tp_{water}} + Unc_{tp_{ice}}}, \tag{5}$$

where $Unc_{IST}$, $Unc_{tp_{water}}$ and $Unc_{tp_{ice}}$ are the uncertainty contributions of $IST$, $tp_{water}$ and $tp_{ice}$, respectively. They are given by

$$Unc_{IST} = \left( \frac{1}{tp_{ice} - tp_{water}} \right)^2 * \sigma_{IST}^2 \tag{6}$$

$$Unc_{tp_{water}} = \left( \frac{IST - tp_{ice}}{(tp_{ice} - tp_{water})^2} \right)^2 * \sigma_{tp_{water}}^2 \tag{7}$$

$$Unc_{tp_{ice}} = \left( \frac{tp_{water} - IST}{(tp_{ice} - tp_{water})^2} \right)^2 * \sigma_{tp_{ice}}^2, \tag{8}$$

where $\sigma_{IST}$, $\sigma_{tp_{water}}$ and $\sigma_{tp_{ice}}$ are the uncertainties of $IST$, $tp_{water}$ and $tp_{ice}$, respectively. We assume that $\sigma_{IST}$ and $\sigma_{tp_{water}}$ are equal to the measurement uncertainty of the MODIS IST, which amounts to 1.3 K [10]. For $\sigma_{tp_{ice}}$, we assume that it is equal to the standard deviation of the 48 iterations done for the retrieval of $tp_{ice}$ in Section 2.4. It is typically between 0.1 and 0.6 K (see Figure 4d).

#### 2.6.2. Merged Sea-Ice Concentration Uncertainty

If MODIS data are available, the first case in Equation (4) needs to be considered, and the merged SIC at the pixel with the coordinates (x,y) is given by

$$SIC_{merged_{x,y}} = SIC_{MODIS_{x,y}} + SIC_{AMSR2_{5\,km}} - SIC_{MODIS_{5\,km}} \tag{9}$$

$SIC_{MODIS_{x,y}}$ and $SIC_{MODIS_{5\,km}}$ are basically two samples of the same dataset, only that $SIC_{MODIS_{5\,km}}$ is the average MODIS SIC in the 5 km by 5 km around the pixel at the coordinates (x,y). We thus consider Equation (9) to be a linear combination of the MODIS and AMSR2 SIC and calculate the uncertainty as

$$\sigma_{SIC_{merged}} = \sqrt{Unc_{SIC_{MODIS}} + Unc_{SIC_{AMSR2}}} \times \frac{1}{\sqrt{2}}, \tag{10}$$

where $\sigma_{SIC_{merged}}$ is the uncertainty of the merged SIC and $Unc_{SIC_{MODIS}}$ and $Unc_{SIC_{AMSR2}}$ are the uncertainty contributions of the MODIS SIC and the AMSR2 SIC, respectively. As $SIC_{MODIS}$ and $SIC_{AMSR2}$ are retrieved from different wavelength regimes and by different approaches, we consider them to be independent observations of the same quantity and thus scale the uncertainty by $\sqrt{2}$. $Unc_{SIC_{MODIS}}$ and $Unc_{SIC_{AMSR2}}$ are given by

$$Unc_{SIC_{MODIS}} = \sigma_{SIC_{MODIS}}^2 \tag{11}$$

$$Unc_{SIC_{AMSR2}} = \sigma_{SIC_{AMSR2}}^2, \tag{12}$$

where $\sigma_{SIC_{MODIS}}$ is the MODIS SIC uncertainty as given by Equation (5) and $\sigma_{SIC_{AMSR2}}$ is the uncertainty of the AMSR2 SIC. We adopt the approach of the authors of [17], who give $\sigma_{SIC_{AMSR2}}$ as a function of the AMSR2 SIC as shown in Figure 2.

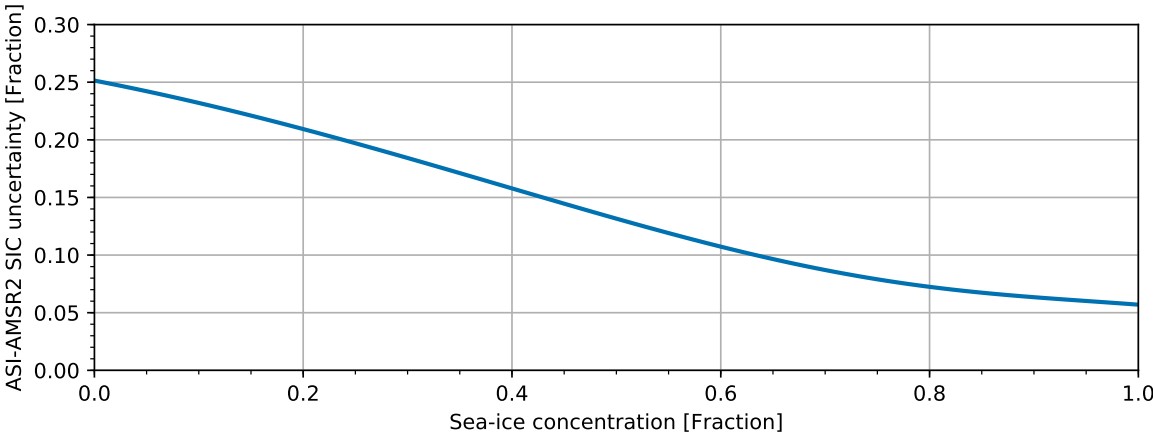

**Figure 2.** Uncertainty of the AMSR2 SIC as derived in [17].

If MODIS data are not available for the merging, the merged SIC uncertainty is equal to the AMSR2 SIC uncertainty.

*2.7. Sentinel-2 Sea-Ice Concentration*

We classify the Sentinel-2 reflectances according to their frequency distribution (see the example in Figure 6b). The three bands from the visible spectrum (bands 2–4) are investigated. Histograms of all 3 bands (red/640 nm, green/560 nm and blue/490 nm) are evaluated for all 79 Sentinel-2 scenes. They show no notable difference for our purpose; all three bands enable a clear distinction between the thin and thick-ice classes if both classes are present and a clear distinction between ice and water if only thick ice is present. We decide to use band 4 (red/640 nm). The reflectances are classified into three classes: open water, thin ice and thick ice. As the ice peaks differ from scene to scene, we select specific thresholds for each scene instead of using a global threshold for each scene. If there are two peaks at the high end of the reflectance spectrum, both are treated as ice. We create two reference SIC datasets: One for which the thin-ice class is treated as ice and one for which it is treated as water. The SIC dataset where the thin ice is treated as water will be called thick-ice SIC as it comprises only thick ice. Consequently, the SIC dataset for which also thin ice is treated as ice will be called thin-ice dataset as it comprises thin and thick ice. The resolution of the binarised ice/water maps is 10 m and the desired resolution of the SIC dataset is 1 km. Thus, we assume that one SIC pixel (1 km) comprises 100 by 100 reflectance pixels. We select each pixel in the SIC array and assign it with the average of the surrounding 100 by 100 reflectance pixels, assuming that the SIC pixel is in the centre of the box. In this way, we obtain a SIC dataset at 1 km resolution from the 10 m reflectance dataset.

## 3. Results

*3.1. MODIS Sea-Ice Concentration*

Analysing the MODIS ice tie-point time-series (Figure 3), we notice that a small number of clear-sky pixels often coincides with a high ice tie-point. We explain this by enhanced evaporation, and thus more fog and convective clouds under higher temperatures. High ice tie-points are problematic for the retrieval because they decrease the dynamic range and thus increase the susceptibility to small IST variations. These cloud and fog features often also are patchy, and thus the high ice tie-point is not representative for a larger area. We therefore omit granules with less than 20% clear-sky pixels for this part of our study. This leaves us with 478 granules, which amounts to 47% of all granules. As the discarded granules are mostly cloud-covered anyway, we only discard 20% of the cloud-free pixels. Using the conservative cloud masking (see Section 2.1) would result in 519 available

granules (52%) and also 20% of discarded pixels. The effect of the stricter cloud mask is presented in Figure 3:

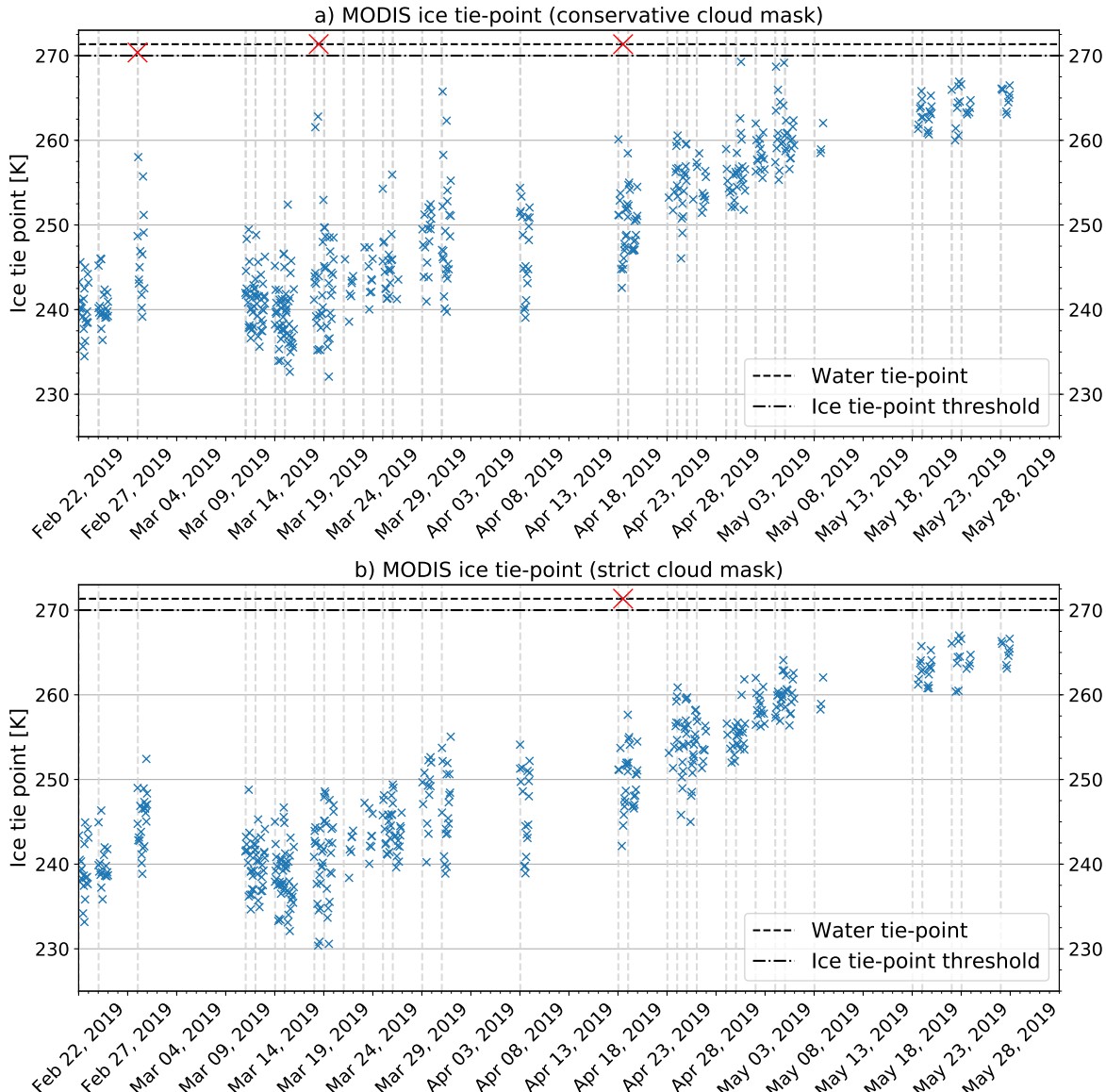

**Figure 3.** Time-series of the ice tie-point when applying the conservative cloud mask (panel (**a**)) and the strict cloud mask (panel (**b**)). The x-axis shows the date; the vertical lines mark the days with Sentinel-2 scenes. The dashed horizontal line shows the water tie-point. The dash-dotted line gives the threshold above which no SIC is retrieved in the current version of our retrieval. Red crosses mark the granules above the threshold.

The time-series of the strictly masked data shows less outliers than that of the conservatively masked data. This happens because the high spikes in the conservatively masked time-series occur when most of the granule is cloud-covered, as described above. Then, a small number of additionally discarded pixels is enough to violate the criteria which must be met for the tie-point retrieval to be performed (see Section 2.4). For example, for the high tie-point on 18 April (red cross in Figures 3 and 4) the retrieval is performed in only one iteration. This also explains why the standard deviation is 0 for this granule (see again Figure 4).

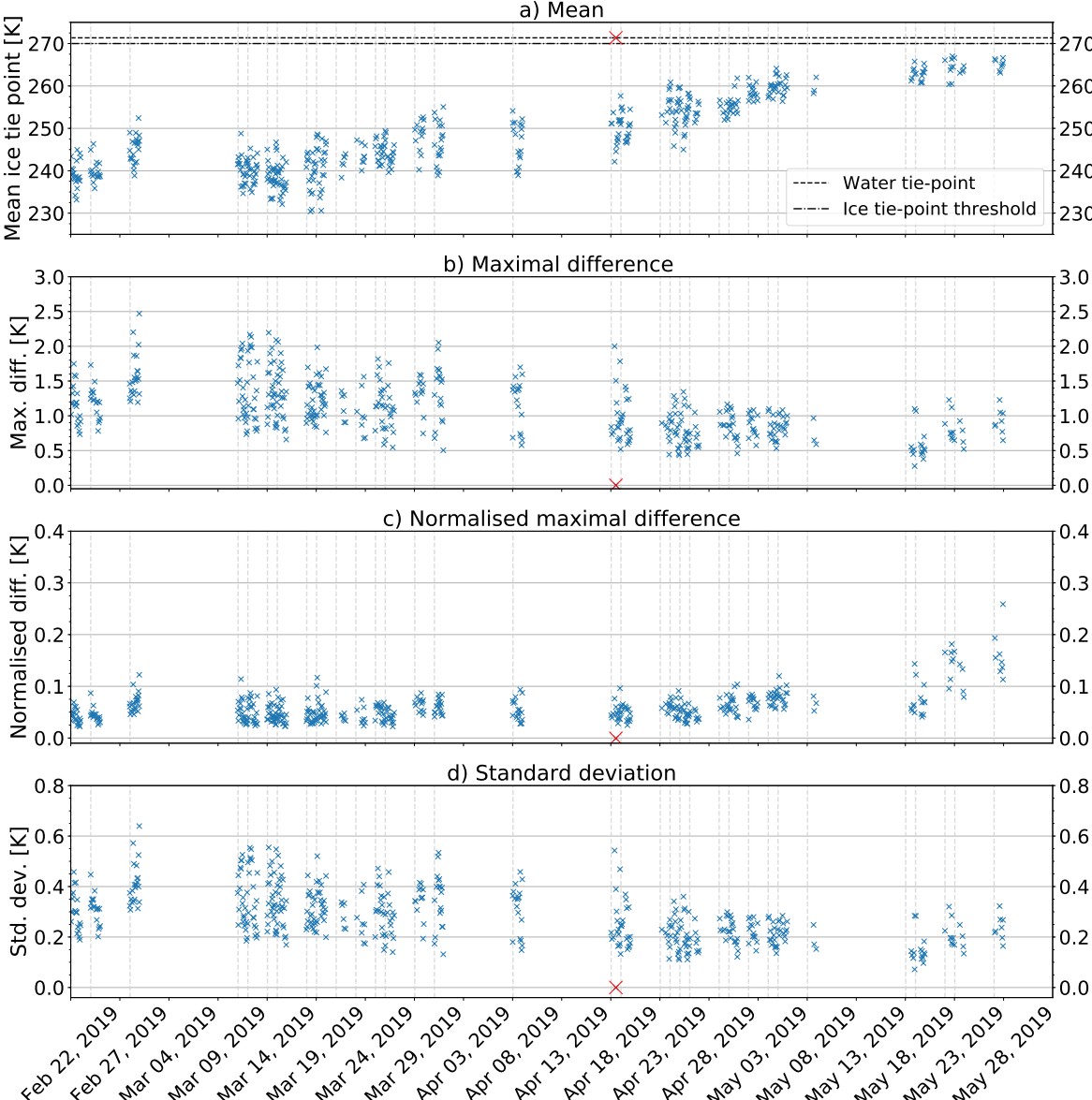

**Figure 4.** Effect of shifting the starting position of the box used for the ice tie-point calculation. The starting position is shifted by one pixel before redoing the calculation, so that each pixel is covered 48 times. The x-axis shows the date; the vertical lines mark the days with Sentinel-2 scenes. Panel (**a**) shows the mean of the 48 iterations, same as in Figure 3b. The dashed horizontal line shows the water tie-point. The dash-dotted line gives a discarding threshold. If the ice tie-point is above this threshold, no SIC is derived in the current version of our retrieval. As in Figure 3, red crosses mark the granules above this threshold. Panel (**b**) shows the difference between the maximal and minimal ice tie-point. Panel (**c**) also shows the maximal difference, but normalised by the dynamic range (difference between ice and water tie-point). Panel (**d**) shows the standard deviation of the 48 iterations, also normalised by the dynamic range.

The number of pixels which are only discarded by the strict cloud mask decreases with time (not shown here). For example, in February, on average 66.4% of a MODIS granule is discarded when applying the conservative cloud mask, while 74.0% of a granule is discarded if the strict cloud masking is applied. This corresponds to a relative increase of 11.5% when applying the stricter cloud mask. In May, on average 81.9% of the MODIS granule is covered by clouds when applying the conservative

cloud mask and 83.7% when applying the strict cloud mask, which means a relative increase of 1.8% by using the stricter cloud mask. We explain this by the limited presence of low, thick clouds in the early months of the year [18,19]. Therefore, the cloud cover is dominated by the presence of thin high clouds whose seasonal cycle is less pronounced [20]. These are more challenging to detect and thus more likely to be flagged "probably clear" or "probably cloudy". These flags are discarded by the stricter cloud mask, but not by the conservative cloud mask. We judge the number of discarded pixels as an acceptable trade-off for the reduced variability and outliers and thus adopt the stricter cloud mask.

In Figure 4, we show the variability of the ice tie-point. This is done by analysing the variation of the 48 possible ice tie-point values per pixel. Ice tie-points on the same day but for different overflights typically differ by 10 K but by up to 20 K in single cases. This reflects the daily cycle and the spatial variability of the IST. Moreover, more than one region is covered on some days. Adjacent granules differ by up to 10 K, which we attribute partly to spatial variability, but mainly to the different cloud cover. Even if the IST distribution in two adjacent granules would be identical, the mean IST would differ depending on which part of the respective granule is obscured by clouds.

The difference between minimal and maximal ice tie-point within one 48 km-by-48 km region is between 1 K and 2 K in February and March and slightly decreases until the end of May. We interpret this a combination of two effects. The first is that under rising temperatures the temperature variability in our 48 km-by-48 km region is smaller than in winter. The second factor is the increased cloud cover. As more pixels are cloud-covered in April and May, the requirements for the tie-point retrieval to be performed are violated more often (see Section 2.4), thus less of the 48 iterations actually yield a possible tie-point and there is a smaller difference between minimal and maximal tie-point.

To assess the influence on our retrieval, we next look into the differences normalised to the dynamic range. This is more meaningful for our purpose as the same difference between minimal and maximal ice tie-point has a larger influence at smaller dynamic ranges and vice versa. Because the ice tie-point increases in May, the normalised difference (Figure 4c) increases from 5% to 20% and more of the dynamic range at the end of May. This is consistent with the assumption that the retrieval yields less reliable results in summer when the ice surface temperature approaches the freezing point. The mean normalised difference of all months is 0.08. This means that choosing an arbitrary starting point for the bilinear regression like it is done in the original retrieval [1] changes the resulting final ice tie-point by up to 8% on average. This is an upper limit of the introduced inaccuracy as it is the difference between the minimum and maximum ice tie-point. The standard deviation of the 48 iterations gives a more representative estimate of the actual inaccuracy introduced by the arbitrary starting point. The mean standard deviation of all 478 granules is 0.33 K. Normalised with respect to the dynamic range, this amounts to 1.7% of the dynamic range. Although the ice tie-point increases in April and May, the standard deviation does not. This means that the accuracy of the approach does not depend on the season until mid May.

Next, we investigate how much this affects the variability of the MODIS SIC. For this, we calculate the MODIS SIC with the final ice tie-point plus/minus one standard deviation and show the resulting SIC together with the difference in Figure 5. The difference increases when the SIC decreases, which we expect because a less compact sea-ice cover is naturally associated with larger variability. The maximal difference is 6.2% and the mean difference amounts to 1.9%. This can be interpreted as the inaccuracy introduced by an arbitrarily chosen starting position for the ice tie-point. Our approach of varying the starting position mitigates this inaccuracy as far as possible.

Our retrieval assumes a constant freezing point, but in reality the freezing point varies depending on salinity. To assess the influence of this, we calculate the MODIS SIC with freezing points of $-1.09\,°C$ (salinity of 20) and $-1.87\,°C$ (salinity of 34). The difference for all granules is on average 0.5% and increase slightly towards the end of May (not shown here).

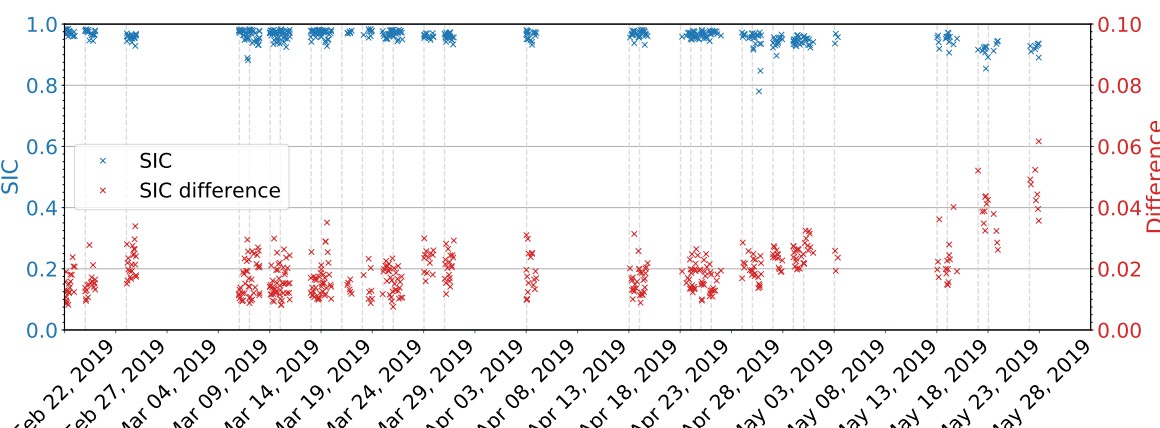

**Figure 5.** MODIS SIC calculated with the ice tie-point from Figure 4a plus/minus the standard deviation from Figure 4d. The blue dots show the SIC. The orange dots show the difference between the MODIS SIC resulting from varying the ice tie-point by plus/minus one standard deviation. The x-axis shows the date; the vertical lines mark the days with Sentinel-2 scenes.

*3.2. Sea-ice Concentration Evaluation*

### 3.2.1. Sentinel-2 Sea-Ice Concentration

Seventy-nine Sentinel-2 scenes are used for constructing the reference Sentinel-2 SIC dataset. Figure 6 shows one example scene from the Kara Sea on 12 March 2019. The RGB image (Figure 6a) shows that there are regions of open water (dark), regions which have been recently overfrozen (greyish) and regions with thicker ice (bright). Treating the thin-ice class as water results in a number of open-water areas within the ice pack for the Sentinel-2 SIC (see Figure 6c). These areas are shown as 100% SIC if the thin ice is treated as ice (Figure 6d). A large open-water lead and an area of smaller ice floes in the north-western corner of the scene are still represented as reduced SIC. The mean of the Sentinel-2 SIC is reduced from 99.6% to 88.1% if the thin ice is included compared to if it is excluded and the standard deviation increased from 3.3% to 23.6%, respectively. Later, we will compare both cases to the merged, MODIS and AMSR2 SIC.

Next, we present the time-series of all Sentinel-2 scenes in Figure 7. The mean thin-ice SIC is 94.4%, while the mean of the thick-ice SIC is 87.3%. The mean SIC standard deviations within one scene are 9.2% (thin-ice SIC) and 13.2% (thick-ice SIC). In scenes with 100% SIC, the thin- and thick-ice SIC is equal, i.e., there is no thin ice. In all other cases, the thin-ice SIC is higher, i.e., there is thin ice.

### 3.2.2. Comparison to MODIS and Merged Sea-Ice Concentration

The MODIS and merged SIC time-series corresponding to the 66 Sentinel-2 scenes with a time lag below two hours are shown in Figure 8. The AMSR2 SIC are not shown as their mean is identical to that of the merged SIC by definition. There is no significant bias between the datasets; the mean merged and MODIS SIC are 93.1% and 92.8%, respectively. The Sentinel-2 thin- and thick-ice SIC means are 94.4% and 87.3%, respectively. The root mean square deviation (RMSD) between the MODIS and merged SIC is 5.0%, which shows that the two datasets indeed differ. In the case with the largest discrepancy, the merged SIC is 94.4% while the MODIS SIC is only 83.3%. Only points where MODIS data are available are considered in the means, so that different coverage can not be the reason. We will later investigate this scene in detail to understand the reasons for the deviation. Generally, the absolute differences to the thin-ice Sentinel-2 SIC are smaller (mostly within ±10%). The absolute differences to the thick-ice SIC range from 10% to −30%, where a positive difference means that the Sentinel-2 SIC are higher. Next, we investigate two scenes in detail: one where the MODIS SIC are of poor quality and one where they are of good quality.

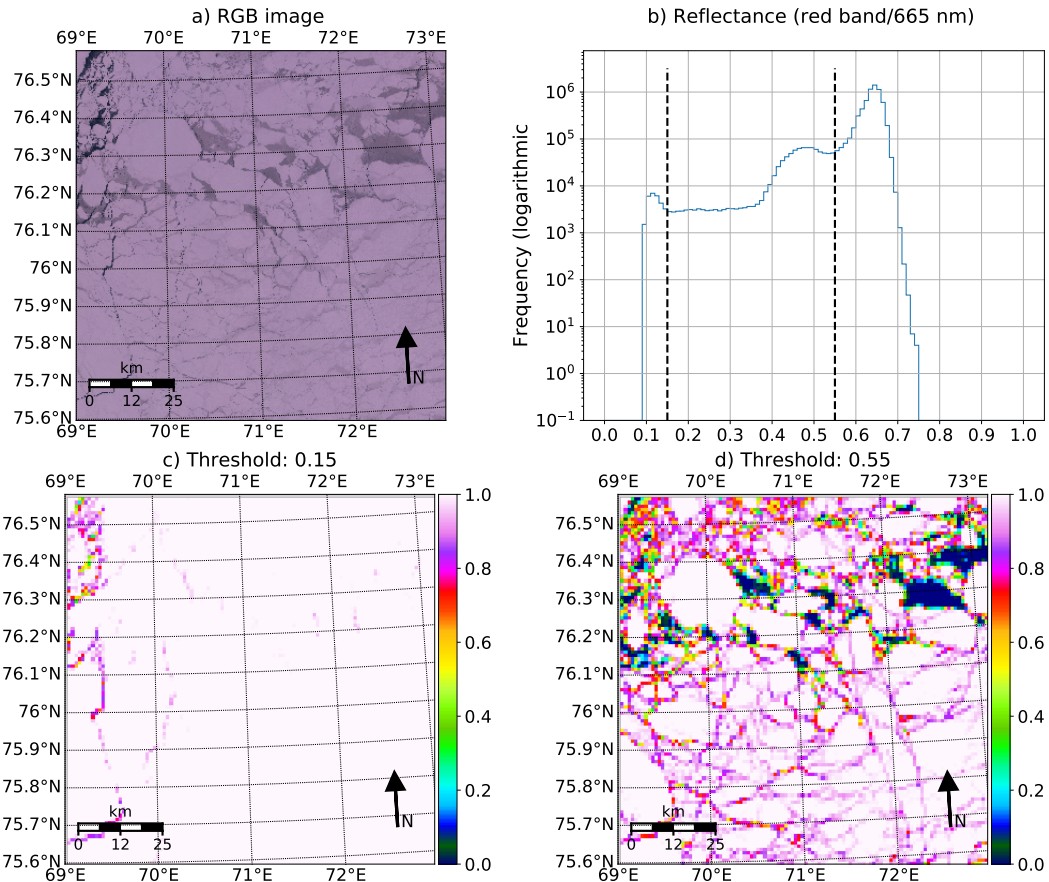

**Figure 6.** Sentinel-2 SIC in the Kara Sea on 12 March 2019. (**a**) RGB image showing open water, thin ice and thicker ice. (**b**) Frequency distribution of the 665 nm reflectance. Mind the logarithmic scaling of the y axis. Vertical dashed lines mark the threshold for the thin and thick-ice class. (**c**) Sentinel-2 SIC if thin ice is treated as ice. (**d**) Sentinel-2 SIC if thin ice is treated as water.

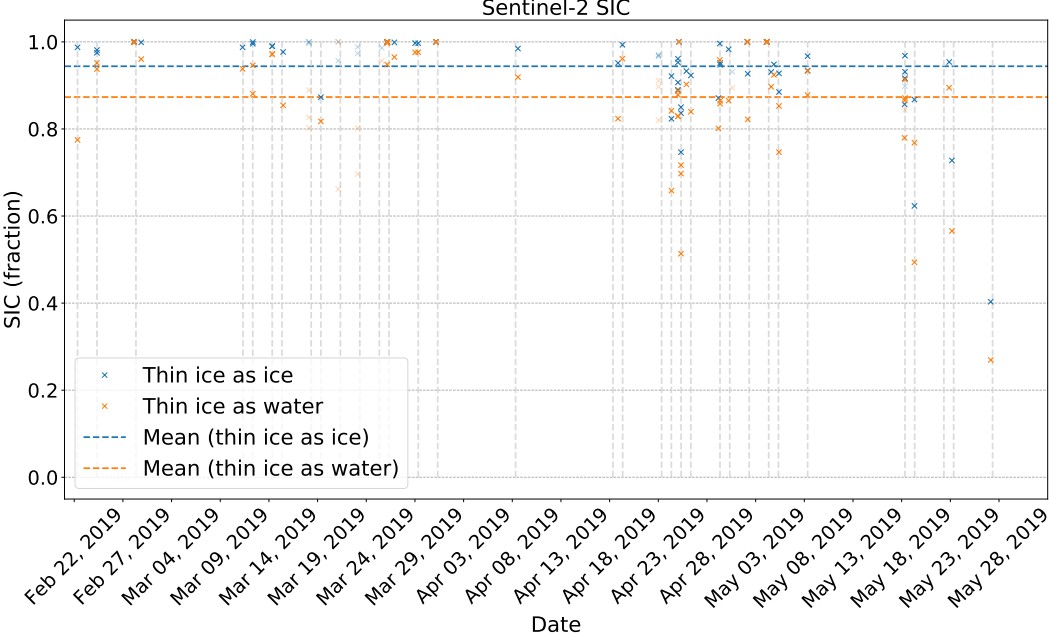

**Figure 7.** Time-series of the mean thin-ice and thick-ice SIC of all Sentinel-2 scenes. The x-axis shows the date, and the vertical lines mark the days with Sentinel-2 scenes. Scenes which are not used for intercomparison with the other SIC datasets because of the time lag exceeding 120 min are indicated by faint markers.

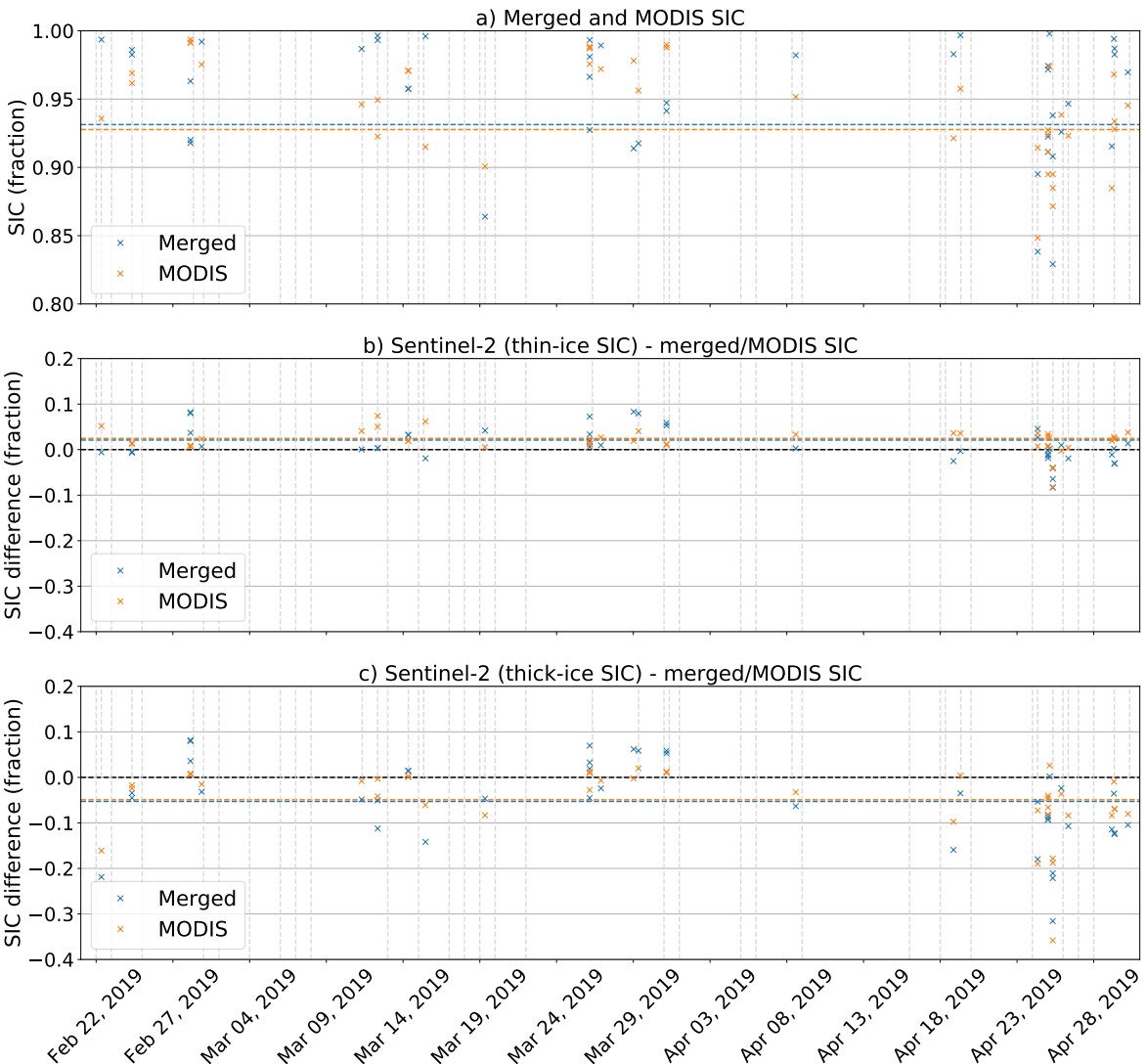

**Figure 8.** (**a**) Time-series of the merged and MODIS SIC corresponding to the Sentinel-2 time-series from Figure 7. The horizontal dashed lines show the mean of the respective dataset. (**b**) Difference between the merged and MODIS SIC and the thin-ice Sentinel-2 SIC from Figure 7. Positive differences mean that the Sentinel-2 SIC is higher. The horizontal dashed lines show the mean difference for the respective dataset. (**c**) Same as panel (**b**), but for the thick-ice Sentinel-2 SIC. The x-axis shows the date, the vertical lines mark the days with Sentinel-2 scenes.

### 3.2.3. Merging with Poor-Quality MODIS Sea-Ice Concentration

In Figure 9, we present SIC maps for the scene with the largest discrepancy between the merged and the MODIS SIC. It is located in the Kara Sea and was acquired on 18 May 2019. The gap in the southwestern corner is not caused by a cloud, but by high MODIS ice tie-points (see Figure 10b,c). The edge of the gap exactly follows the 270 K contour, above which no MODIS SIC are retrieved currently (see Section 2.4). The poor quality of the MODIS SIC all over the scene shows that this threshold which we chose initially is too conservative in this case. Special attention is thus paid in Section 3.3 to derive a threshold above which no MODIS SIC retrieval is executed.

The performance of the MODIS SIC retrieval is further deteriorated by an unscreened cloud in the eastern part of the scene (Figure 10a), so that this scene shows both the susceptibility of the MODIS SIC retrieval towards high surface temperatures and the susceptibility towards unscreened clouds.

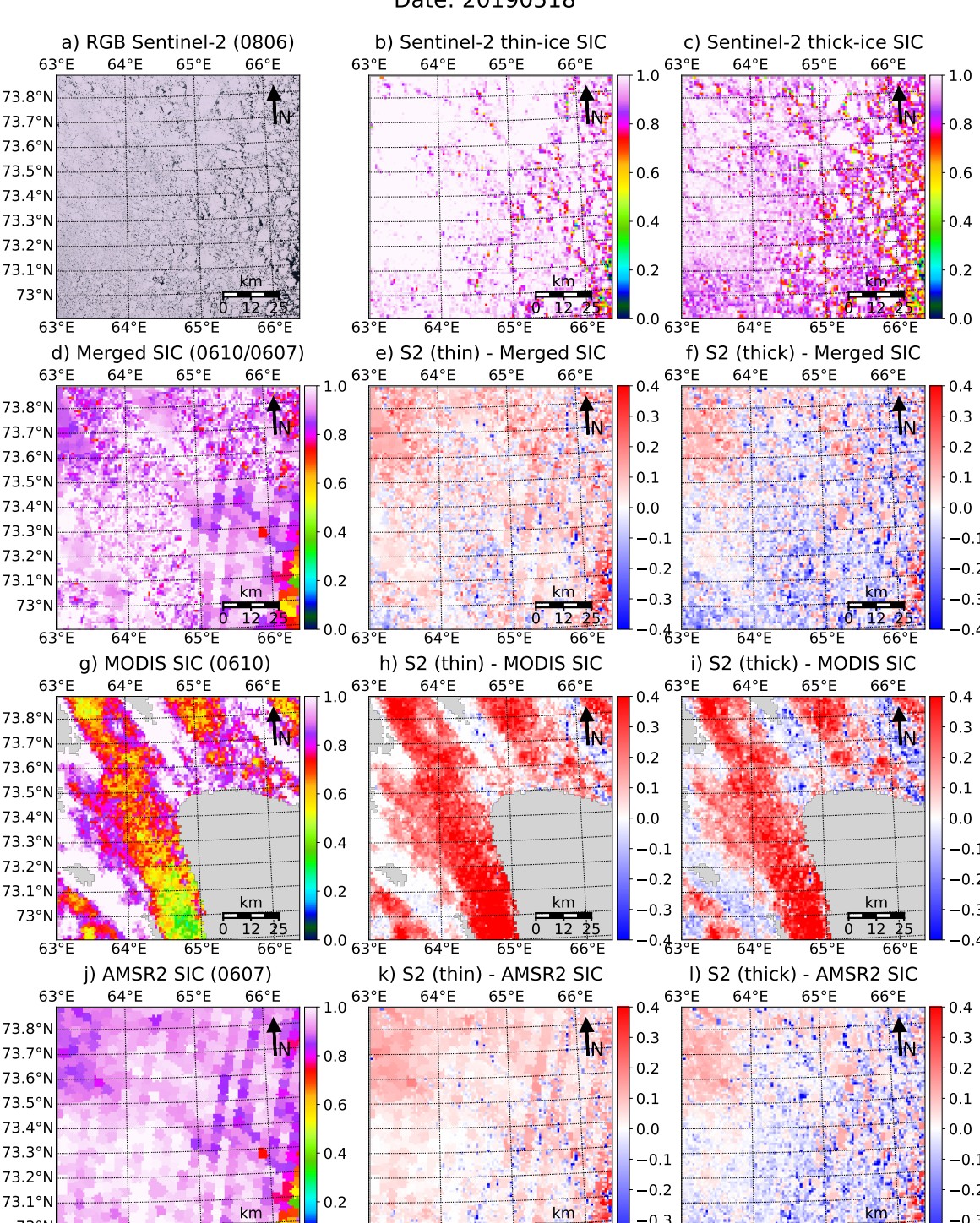

**Figure 9.** Sentinel-2 scene in the Kara Sea on 18 May 2019 and the corresponding SIC maps. (**a**) Sentinel-2 RGB image. Sentinel-2 SIC with thin ice considered as ice (**b**) and water (**c**). (**d**–**f**) The merged SIC and their difference to the thin- and thick ice Sentinel-2 SIC. Panels (**g**–**i**,**j**–**l**) show the same for MODIS and AMSR2 SIC, respectively. The acquisition times for the single scenes in UTC are given in brackets in the titles of the maps in the left row.

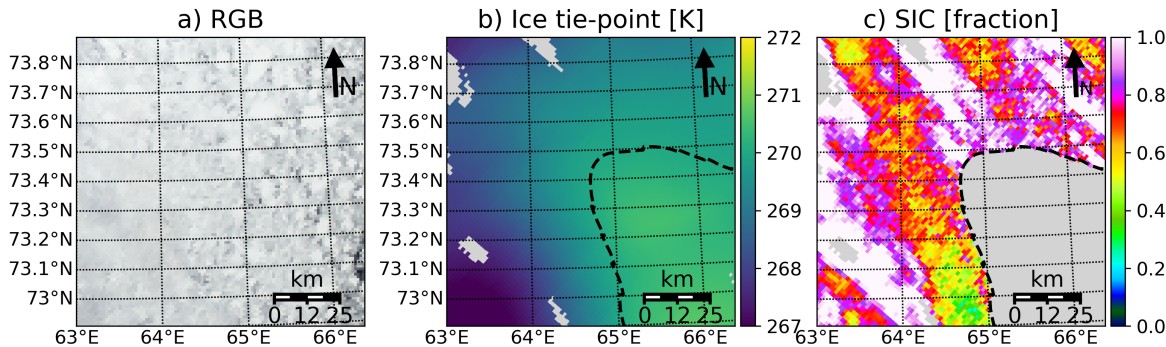

**Figure 10.** (**a**) MODIS RGB image of the scene in Figure 9g. (**b**) Ice tie-point of the MODIS SIC. The dashed line marks the 270 K threshold above which no SIC retrieval is performed. (**c**) MODIS SIC as in Figure 9g.

In the part without MODIS data, the actual resolution of the merged dataset goes down to 5 km. The almost vertical stripes in the AMSR2 and merged SIC are due to rapid fluctuations in the radiometer gain between the two scans of the 89 GHz channels that cannot be corrected for currently. The part of the scene where MODIS data are available is resolved more finely, but the structures do not resemble those of the Sentinel-2 image. The finer resolution patterns of the merged SIC resemble the reference SIC, however the single floes are too small and the time difference is too large (114 min) to recognise distinct floes of the reference SIC scene in the merged SIC. The influence of the large difference between the MODIS SIC and the Sentinel-2 SIC on the merged SIC is mitigated as we tune the merged SIC to preserve the AMSR2 mean. We conclude that in this scene including the poor-quality MODIS SIC does not significantly deteriorate the quality of the merged SIC, but we also cannot conclude that it enhances the quality.

All datasets underestimate the thin-ice SIC; the merged and AMSR2 SIC by 3.6% and 3.8%, respectively; and the MODIS SIC by 16.7%. The difference compared to the thick-ice SIC is smaller for all datasets ($-1.4\%$, 11.7% and $-1.3\%$ for the merged/MODIS/AMSR2 SIC, respectively). The percentages are absolute values, i.e., not relative to the other percentage value. The small differences do not necessarily indicate a more precise result, but are the consequence of positive and negative differences cancelling out for the merged and AMSR2 SIC. This is underlined by their RSMD. It is smaller when computed towards the thin-ice SIC (0.093 for the merged SIC and 0.069 for the AMSR2 SIC) than when computed towards to the thick-ice SIC (0.110 for the merged SIC and 0.092 for the AMSR2 SIC). For the MODIS SIC, the RMSD is lower for the thick-ice SIC. To demonstrate the benefit of the higher resolution, we compare the open-water extent (area of pixels covered by at least 15% of water) of the datasets. We only consider pixels where all datasets are available, which yields an area of 8191 km$^2$. The open-water extent of the thin- and thick-ice SIC is 208 km$^2$ and 1045 km$^2$, respectively. The open-water extent of the merged SIC (818 km$^2$) is close to the thick-ice open-water extent (1045 km$^2$), but higher than the thin-ice open-water extent (208 km$^2$). The AMSR2 open-water extent (292 km$^2$) is close to the thin-ice open-water extent, but lower than the thick-ice open-water extent. The MODIS open-water extent of 4508 km$^2$ is higher than both the thin- and the thick-ice open-water extent. An overview over all parameters is given in Table 2.

**Table 2.** Comparison of the different SIC datasets in Figure 9. We show the mean μ, the mean difference Δ between the thin- and thick-ice Sentinel-2 SIC and the respective dataset and the RMSD with regard to the thin- and thick-ice Sentinel-2 SIC and the open water extent (area of pixels with less than 85% SIC). A positive difference means that the Sentinel-2 SIC is higher. All quantities except the open water extent are given as fraction of 1. Only pixels where all datasets are available are considered for the comparison. For this scene, this amounts to 8191 of 11,881 pixels, equivalent to an area of 8191 km². Details about location and acquisition time of the scene are given in the caption of Figure 9.

|  | **Merged** | **MODIS** | **AMSR2** | **S2 (Thin)** | **S2 (Thick)** |
|---|---|---|---|---|---|
| μ | 0.944 | 0.813 | 0.943 | 0.981 | 0.930 |
| $\Delta_{thin}$ | 0.036 | 0.167 | 0.038 | N/A | N/A |
| $\Delta_{thick}$ | −0.014 | 0.117 | −0.013 | N/A | N/A |
| $RMSD_{thin}$ | 0.093 | 0.235 | 0.069 | N/A | N/A |
| $RMSD_{thick}$ | 0.110 | 0.212 | 0.092 | N/A | N/A |
| $OWE$ [km²] | 818 | 4508 | 292 | 208 | 1045 |

### 3.2.4. Merging with Good-Quality MODIS Sea-Ice Concentration

An example to demonstrate the benefit of including good-quality MODIS SIC is presented in Figure 11, which shows a scene recorded in the Beaufort Sea on 22 May 2020. The scene comprises one large open lead, several narrower leads and large, contiguous ice floes of 100% SIC. Some of the narrower leads are completely overfrozen, some are partly overfrozen and some are completely open. It is hard to determine one scene-wide threshold which classifies all pixels with open water as open and all overfrozen leads, even those with frazil or very thin ice, as overfrozen. Some misclassification is likely to occur, so that the thin-ice SIC probably also includes small amounts of pixels which are classified as ice, but are actually open water. Our approach of choosing one thin-ice threshold per scene, not one global thin-ice threshold for all scenes, reduces the risk of this misclassification.

The MODIS SIC resolve the leads as reduced SIC, but undershoot the reference SIC of the larger ice floes. The AMSR2 SIC consistently show 100% SIC over the large floes. The overfrozen yet still recognisable leads are not visible in the AMSR2 SIC, except for a coarsely resolved large open lead in the middle of the scene. This is not a failure of the algorithm. Rather, the coarse resolution is because of the frequency and the insensitivity towards overfrozen leads is because the polarisation difference which the algorithm uses to derive SIC approaches the AMSR2 ice tie-point rapidly once the freezing started [4]. Furthermore, the nonlinear nature of the AMSR2 algorithm reduces variability near 100%. This can lead to an overestimation of SIC between 90% and 100% and may explain why we do not see traces of narrow leads in the AMSR2 SIC data. The merged SIC resolves both the narrow, overfrozen leads and the large, open lead as reduced SIC. At the same time, tuning them to preserve the AMSR2 mean causes them to show close to 100% SIC over the large ice floes where the MODIS SIC are around 90%. A small open-water area in the southeastern corner of the scene is partly masked in the MODIS SIC, but retrieved by the AMSR2 and the merged SIC.

The mean of the merged SIC is 98.7%, while the means of the thin- and thick-ice Sentinel-2 SIC are 95.9% and 90.2%, respectively. The MODIS SIC is 88.7%, which means an underestimation compared to both the thin- and the thick-ice Sentinel-2 SIC. The AMSR2 SIC have the smallest RMSD (12.0%) compared to the thin-ice SIC; the RMSDs of the merged SIC (13.5%) and the MODIS SIC (15.6%) are higher. All RMSDs increase when considering thin ice as water (thick-ice SIC). The MODIS RMSD is now 16.3%, while the merged and AMSR2 RMSDs are at 19.9% and 19.8%, respectively. The open-water extent of the merged SIC (771 km²) is close to that of the thin-ice Sentinel-2 SIC (878 km²). The open-water extent of AMSR2 is much smaller (182 km²), while that of MODIS is much higher (2760 km²). This value is close to the thick-ice Sentinel-2 open-water extent (2313 km²). The area where measurements of all three sensors are available is 11,350 km². Table 3 gives an overview over all parameters.

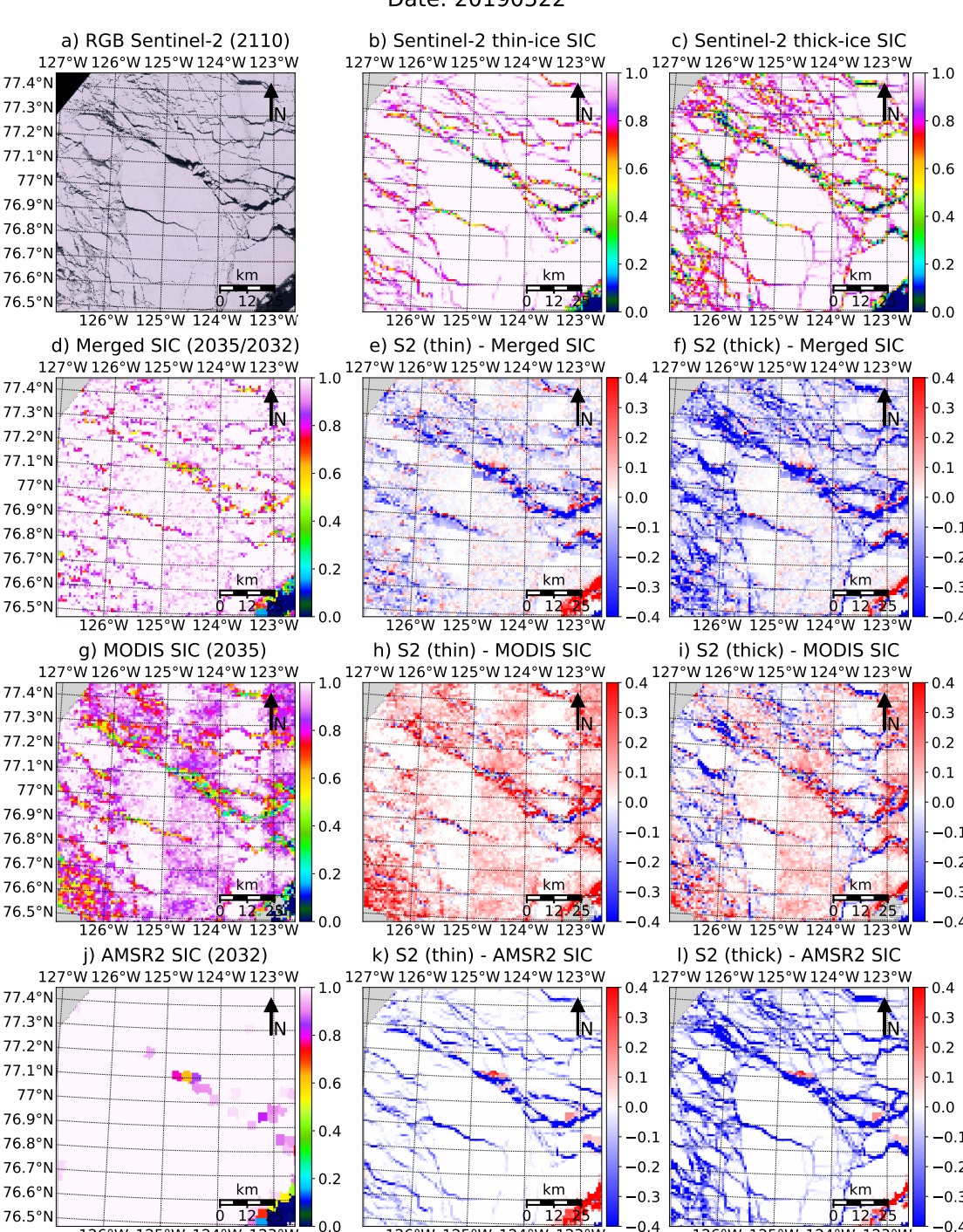

**Figure 11.** Sentinel-2 scene in the Beaufort Sea on 22 May 2019 and the corresponding SIC maps. (**a**) Sentinel-2 RGB image. Sentinel-2 SIC with thin ice considered as ice (**b**) and water (**c**). (**d–f**) The merged SIC and their difference to the thin- and thick ice Sentinel-2 SIC. Panels (**g–l**) show the same for the MODIS and AMSR2 SIC, respectively. The acquisition times for the single scenes in UTC are given in brackets in the titles of the maps in the left row.

**Table 3.** Like Table 2, but for the data from Figure 11. All datasets are available for 11,350 out of 11,881 pixels, which yields a surface area of 11,350 km$^2$.

|  | Merged | MODIS | AMSR2 | S2 (Thin) | S2 (Thick) |
|---|---|---|---|---|---|
| $\mu$ | 0.987 | 0.887 | 0.988 | 0.959 | 0.902 |
| $\Delta_{thin}$ | $-0.028$ | 0.072 | $-0.028$ | N/A | N/A |
| $\Delta_{thick}$ | $-0.085$ | 0.015 | $-0.085$ | N/A | N/A |
| $RMSD_{thin}$ | 0.135 | 0.156 | 0.120 | N/A | N/A |
| $RMSD_{thick}$ | 0.199 | 0.163 | 0.198 | N/A | N/A |
| $OWE\,[\text{km}^2]$ | 771 | 2760 | 182 | 878 | 2313 |

Thus, in this case we conclude that the merged SIC shows clear advantages over the AMSR2 SIC because of its higher resolution and ability to resolve almost all leads identified in the Sentinel-2 reference SIC. Moreover, compared to the MODIS SIC, the merged SIC shows clear advantages because the bias and RMSD compared to the reference SIC are lower and unnatural fluctuations in the MODIS SIC are removed.

*3.3. Sea-Ice Concentration Uncertainties*

3.3.1. MODIS Ice Tie-Point Threshold

The MODIS SIC uncertainties (Equations (5)–(7)) become large if the ice tie-point is close to the water tie-point. This can even lead to uncertainties which are larger than 1 in single cases. We consider a MODIS SIC with an uncertainty of 0.3 (i.e., 30%) as the upper boundary for a meaningful SIC. This is much larger than what one would accept as uncertainty for the final merged dataset. However, because the uncertainty of the merged SIC gets reduced we accept higher uncertainties for the MODIS SIC because they still can provide valuable spatial variability. As the water tie-point is fixed, we need to identify an ice tie-point threshold above which meaningful SIC cannot be derived any more. To find this threshold, we investigate the uncertainties for all 997 MODIS granules for which we have Sentinel-2 scenes. We show the mean and the 99th percentile of the uncertainties for each granule in Figure 12. The parameters are shown for IST thresholds between 265 and 270 K, where ice tie-point values above the respective threshold are discarded.

We learn from Figure 12 that the highest threshold, 270 K, would yield mean uncertainties above 0.3 in 69 cases, corresponding to 6.9% of all cases. A threshold of 266.5 K or lower would have to be chosen to discard all granules with a mean uncertainty above 0.3. The time-series of the 99th percentile (Figure 12c) shows that the 99th percentile of the uncertainties exceeds our sanity threshold of 0.3 for 113 granules in May (44.7% of the May granules), and for 169 granules before May (22.7% of the granules before May) when applying a threshold of 269 K or higher. A threshold of 266.5 K yields 99th percentile values of above 0.3 for 32 granules before May (4.3% of all granules before May) and for 31 granules in May (12.3% of the May granules). The mean uncertainty does not show a trend until the end of April and then increase slightly, regardless of the threshold which is chosen. Except for some outliers, the mean uncertainties do not depend strongly on which threshold is chosen. This changes towards the end of the time-series, when the higher thresholds exhibit a higher variability.

The final threshold will be a compromise between reducing the uncertainty as far as possible and, at the same time, keeping as many pixels as possible. Therefore, we also investigated how many pixels would be discarded by each threshold (Figure 12d). We present the fraction of discarded pixels which would be discarded additionally to the pixels that are discarded by the highest threshold anyway. All percentages are absolute values, i.e., an increase of 10% means an increase from, for example, 50% to 60%, not from 50% to 50.5%. Figure 12d shows that applying the lowest threshold, 265 K, would result in discarding up to 30.0% more pixels relative to the number which is discarded by the highest threshold of 270 K. However, on average (see Table 4), only 1.2% of the pixels would be discarded additionally by the lowest threshold. As expected, the number of additionally discarded

pixels increases strongly in May due to the enhanced presence of clouds. This shows that our retrieval is expected to perform less reliably in May than in the other months, but we can be confident that the performance will be stable at least until the end of April.

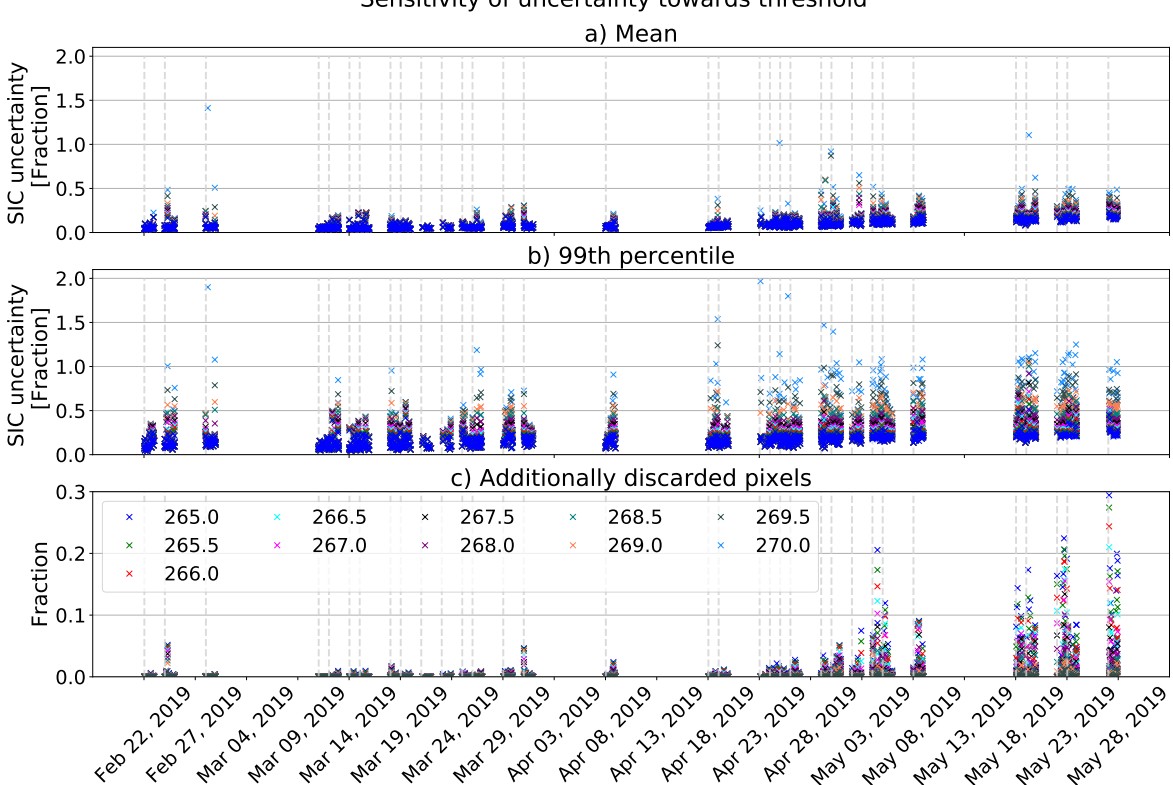

**Figure 12.** Uncertainty of MODIS SIC per granule. The panels show the mean (**a**) and the 99th percentile (**b**) for each granule. Panel (**c**) shows the "additionally discarded values", i.e., the absolute difference between the fraction of values which would be discarded by the respective threshold and the fraction of values would be discarded by the highest threshold. The markers in the panels show the respective parameter for different thresholds between 265 and 270 K, where ice tie-point values above the respective threshold are discarded. The x-axis shows the date, the vertical lines mark the days with Sentinel-2 scenes.

**Table 4.** Number of additionally discarded pixels for each threshold, normalised to the number of pixels in one MODIS granule. "Additionally discarded" means the increase in the number of discarded pixels compared to the number of pixels which the highest (i.e., most conservative) threshold, 270 K, would have discarded. The numbers are absolute values, i.e., an additionally discarded fraction of 0.003 for the lowest threshold, 265 K, in February means an increase from 0.772 (77.2% of all pixels) to 0.775 (77.5% of all pixels). For the highest threshold of 270 K, the absolute of discarded pixels are given.

| Threshold | 265.0 | 265.5 | 266.0 | 266.5 | 267.0 | 267.5 | 268.0 | 268.5 | 269.0 | 269.5 | 270 |
|---|---|---|---|---|---|---|---|---|---|---|---|
| $n_{feb}$ | 0.003 | 0.003 | 0.003 | 0.002 | 0.002 | 0.002 | 0.001 | 0.001 | 0.001 | 0.000 | 0.772 |
| $n_{mar}$ | 0.013 | 0.011 | 0.009 | 0.007 | 0.006 | 0.004 | 0.003 | 0.002 | 0.001 | 0.001 | 0.772 |
| $n_{apr}$ | 0.003 | 0.002 | 0.002 | 0.001 | 0.001 | 0.001 | 0.000 | 0.000 | 0.000 | 0.000 | 0.738 |
| $n_{may}$ | 0.030 | 0.026 | 0.022 | 0.018 | 0.014 | 0.010 | 0.007 | 0.005 | 0.003 | 0.001 | 0.826 |
| $n_{all}$ | 0.012 | 0.010 | 0.009 | 0.007 | 0.006 | 0.004 | 0.003 | 0.002 | 0.001 | 0.001 | 0.776 |

In summary, Figure 12 shows that a threshold of 266.5 K or lower needs to be chosen to ensure that the mean uncertainty is below our sanity threshold of 0.3 for all granules. At the same time, thresholds lower than 266.5 K would lead to discarding a high number (15–30%) of pixels in May. We thus adopt a threshold of 266.5 K.

### 3.3.2. Uncertainty Time Series

In Figure 13, we show the mean merged, AMSR2 and MODIS SIC uncertainties for the time between February 22nd and May 27th. Only pixels where MODIS and AMSR2 data are available are considered. The MODIS SIC uncertainty increases with time as the main uncertainty contributor, the dynamic range, decreases. It is important to note that the dynamic range starts to decrease in April (see Figure 4, an increase in ice tie-point means a decrease in dynamic range), while the MODIS SIC uncertainty only increases in May. We explain this by the nonlinear increase of the single MODIS SIC uncertainty components (see Equations (6) and (7)) for a linearly decreasing dynamic range. Therefore, a decrease of the dynamic range by for example 1 K has a stronger effect on the uncertainty in May, already starting at a smaller dynamic range than in April.

**Figure 13.** The mean uncertainty of the merged, AMSR2 and MODIS SIC is shown in panel (**a**). Panel (**b**) shows the difference between the MODIS SIC and the merged SIC uncertainty, the monthly mean difference is shown as solid horizontal line. The uncertainty is given as a fraction of 1. The x-axis shows the date, the vertical lines mark the days with Sentinel-2 scenes.

The increase of the MODIS SIC uncertainty also causes the uncertainty of the merged SIC to increase with time. In fact, the temporal evolution of the MODIS and merged SIC is nearly identical as the AMSR2 SIC is quite constantly between 6% and 8% over the entire period. Again, note the much simpler assumption for the AMSR2 SIC uncertainty we are applying, which, e.g., does not take atmospheric variability into account. Therefore, the temporal variability of the MODIS SIC directly propagates into the merged SIC. The comparably constant uncertainty of the AMSR2 SIC is because our selection of Sentinel-2 scenes comprises mainly scenes with high SIC and the AMSR2 uncertainty is expressed as a function of SIC. It should therefore not be interpreted as an assessment of the AMSR2 uncertainty under different conditions.

The benefit of the merged dataset over the MODIS SIC with regard to uncertainties is best demonstrated by the difference between their respective uncertainties. Figure 13b shows that the uncertainty is reduced in most cases by including the AMSR2 data. Until April, this is mostly reflected in single scenes in which the MODIS SIC uncertainty is between 10% and 18%, while the merged SIC uncertainty is between 8% and 14%. Apart from these single scenes, the MODIS SIC uncertainty is mostly between 5% and 8%, while the merged SIC uncertainty is between 5% and 6%. In May, the uncertainties of the merged and MODIS SIC start to increase. The increase of the merged SIC uncertainty is less pronounced than that of the MODIS SIC uncertainty. This means that the effect

of including the AMSR2 data reduces the uncertainties stronger for higher uncertainties. By the end of May, the mean uncertainty of the merged SIC is 18%, while the MODIS SIC uncertainty is 25%. When interpreting these numbers, the reader should keep in mind that there are large scene-to-scene fluctuations of up to 10% (merged SIC uncertainty) and up to 15% (MODIS SIC uncertainty). Considering the entire time-series, the uncertainty of the merged dataset is 8.9%, while that of the MODIS SIC is 10.6%. If only May is considered the mean uncertainties of the merged and the MODIS SIC are 11.6% and 14.9%.

### 3.3.3. Uncertainty Evaluation

We evaluate the uncertainties by comparing them to the differences towards the Sentinel-2 reference SIC from Figure 8. Figure 14 shows that in the majority of cases, our uncertainty estimate is larger than the actual difference to the reference Sentinel-2 SIC. The uncertainties for the merged SIC are in most cases around 6%, go up to 15% and reach higher values only in exceptional cases. The differences are mostly close to 0% and hardly above 10%. In 79.2% of the cases, the uncertainty is higher than the difference. The uncertainties of the MODIS SIC are higher than those of the merged SIC. Uncertainties up to 20% occur frequently, but the majority is close to 5% and in 84.8% of the cases the uncertainty is higher than the difference. The AMSR2 SIC uncertainties are centred at about 5% and hardly exceed 8%. Here, our uncertainty estimates are higher than the differences in 72.8% of the cases. We conclude that the uncertainties are a conservative assumption of the actual difference to the reference SIC. In the majority of cases ($\approx$80%) the actual deviation from the real SIC should be lower than the uncertainty provided in the merged SIC dataset.

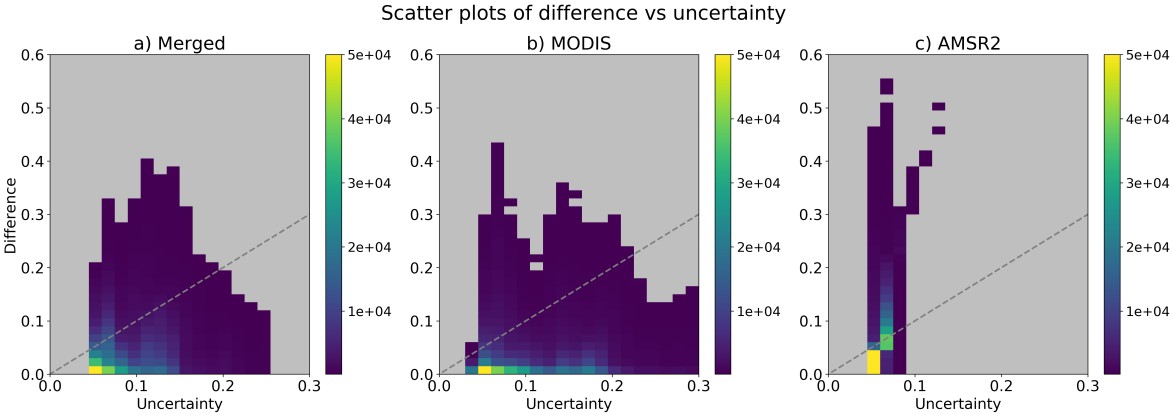

**Figure 14.** Density scatter plot of the absolute difference to the thin-ice Sentinel-2 SIC against the merged, MODIS and AMSR2 SIC uncertainty in panels **(a)**, textbf(b) and **(c)**, respectively. All pixels, i.e., each pixel from each scene used for the time-series in Figure 8 are shown. We exclude pixels where one dataset is not available from all other datasets as well. This results in 743,202 pixels. We use 40 bins per dimension and limit the data to the range between 0 and 0.6. All difference/uncertainty pairs which occur less than 150 times are excluded. The colour scale is cut at 50,000.

## 4. Discussion

The quality of the MODIS SIC which we present in Section 3.1 depends crucially on the reliability of the cloud screening. Specifically, the MODIS cloud mask does not always identify thin clouds and fog correctly [10]. We account for this by masking clouds strictly by only tolerating pixels which are flagged "confident clear". This means that the confidence in the pixel being cloud-free is more than 99% [21]. Even better cloud screening might be achieved with the Fuzzy Cloud Artifact Filter proposed by the authors of [22], but using the MODIS cloud mask alone; this is the best we can do for cloud screening. Additionally, the merging procedure makes sure that even if there are unscreened clouds, the quality of the merged SIC is not much worse than that of the AMSR2 SIC, as we show in Section 3.2 and Figure 9.

Calculating the MODIS SIC, we find that changing the freezing point from $-1.87\,°C$ to $-1.09\,°C$ (corresponding to salinities of 34 and 20, respectively) only introduces a mean difference of 0.5%. A salinity of 20 is the lower limit of what one would expect for the marginal seas [23]. It is higher in the Central Arctic, so that the small mean difference of 0.5% is already an upper limit of the actual error. The authors of [15] find an even lower sensitivity of 0.2%. We thus consider the error introduced by the assumption of a constant freezing point as negligible. Additionally, not retrieving SIC for ice tie-points higher than 266.5 K in future makes sure that the dynamic range is large enough to be robust against small freezing point variations.

Both MODIS SIC and AMSR2 SIC are influenced by sea-ice thickness. The AMSR2 SIC underestimate a SIC of 100% by up to 50% for very thin ice (thinner than 6 cm), but are close to 100% SIC for ice thicker than this under the surface conditions of our study [4]. The authors of [5] confirm this by reporting SIC underestimations of 5% or less for sea-ice thicker than 10 cm. Other passive microwave SIC algorithms are affected more strongly by sea-ice thickness [4,5].

The MODIS SIC underestimation due to sea-ice thickness is not linked to a certain thickness. Instead, it depends on the distribution of sea-ice thickness. The MODIS SIC algorithm assumes a bimodal ice thickness distribution within the region used for the ice tie-point retrieval. If more than one sea-ice thickness class is present in this region, the ice tie-point will represent the ice-surface temperature above the thicker ice, while the thin ice will appear warmer due to the oceanic heat flux and will thus be retrieved as reduced SIC. This is desired by the authors of [1] as their primary goal is to represent the thermal forcing of the atmosphere rather than the actual fraction of ice. We want to retrieve this fraction and thus tune the MODIS SIC to the mean of the AMSR2 SIC.

A situation with inhomogeneously distributed sea-ice thickness as described above can happen throughout the winter season, while sea ice which is thin enough to appear as reduced SIC in the AMSR2 SIC is not expected that often, especially in winter then the large temperature contrast between ocean and air lets the ice grow quickly. We thus consider the AMSR2 SIC to be less dependent on sea-ice thickness than the MODIS SIC, which is again an argument for tuning the MODIS SIC to match the mean of the AMSR2 SIC.

The main benefit of including the MODIS data is the higher potential for identifying leads. Generally, it is debatable whether an overfrozen lead should be shown as reduced SIC, as strictly speaking the SIC should be 100% instantly when the lead starts to refreeze. For most applications, e.g., heat flux calculation or navigation, the presence of leads is very relevant. A calculation in [24] shows that for thick ice the heat flux would be ~9 Wm$^{-2}$ for 99% SIC and 5 Wm$^{-2}$ for 100% SIC. In a 1 km grid cell of our dataset, this would correspond to a difference of 4 MW. We see this and the importance of leads for navigation as a sufficient argument for keeping the leads as reduced SIC in our merged dataset. The benefit of keeping the leads is also reflected in the higher open-water extent of the merged and MODIS SIC compared to the AMSR2 SIC.

For lead retrieval only, the MODIS SIC would be more suitable than the merged SIC. However, our main goal is to have a dataset which does not only allow lead identification, but also offers spatial continuity. We therefore see our dataset as a compromise where a part of the lead information from the MODIS data is lost, but we judge the spatial continuity as valuable enough to accept this as a trade-off. For users who are mainly interested in leads, the MODIS SIC will be provided along with the merged SIC in a future version of our dataset.

Both MODIS and AMSR2 SIC are strongly influenced by melt ponds. We do not consider this relevant for our study here since we do not cover the melt season, but it makes MODIS SIC retrieval in summer impossible, especially if we discard pixels with ice tie-points above 266.5 K as derived in Section 3.3. The merged SIC are thus only available between October and May.

We do not compare our merged SIC to passive microwave SIC. The reason for this is that the main advantage of our product over the well-established passive microwave SIC algorithms is the finer resolution. As the ASI-AMSR2 SIC have the finest spatial resolution of all passive microwave SIC algorithms, we expect the ASI-AMSR2 SIC to yield an open-water extent (the metric which we use to

show the advantages of the merged SIC) which is closer to that of the Sentinel-2 reference SIC than for the other passive microwave algorithms. Showing that the open-water extent of our merged SIC is closer to the Sentinel-2 open-water extent than that of the ASI-AMSR2 SIC does thus immanently show that the merged SIC are also better compared to the other passive microwave SIC with their even coarser spatial resolution. As we preserve the mean of the ASI-AMSR2 SIC, previous comparisons of ASI-AMSR2 SIC to alternative SIC products [24–27] are also valid for our product in terms of potential biases, although the variability might be different.

The selection of Sentinel-2 scenes for evaluation could introduce a bias into our dataset as we are (a) limited to daylight conditions, (b) limited to the marginal seas and (c) limited to temporally unevenly sampled scenes. Our results are thus only representative for these conditions and regions, they should not be interpreted as an assessment of the Arctic-wide SIC transition from late Winter to early Summer. However, the 66 reference scenes cover a fair amount of different ice conditions and a higher than average percentage of intermediate ice concentrations due to their location in the marginal seas. Low and intermediate SIC retrievals are in general more challenging and have higher uncertainties than the close to 100% SIC in the central Arctic. Thus, we do not expect worse results for such cases.

For evaluation, the authors of [15] compare the MODIS SIC with SIC derived from aircraft measurements and find an error of $\pm 10\%$ in April. Comparison with SSM/I data, also done by [15], yields a linear regression error of $\pm 7\%$. The MODIS SIC uncertainties derived by us are on average lower than the ones from the work in [15], but are larger (up to 20%) in some cases. In May, they range from 10% to 35%, i.e., they are larger than the ones in [15]. However, as the values in April, when the work in [15] was done, agree rather well, we conclude that our MODIS SIC uncertainty estimates are in the range of what we expect from the literature.

We make a quite simple approach for estimating the AMSR2 SIC uncertainty by simply assuming them as a function of the SIC after [17], yielding uncertainties between 5% and 10%. The accuracy of the AMSR2 SIC and 12 other passive microwave SIC algorithms has been assessed by [26]. They report accuracies between 3.1% and 8.1% compared to a reference dataset of 75% SIC in winter (Table 2b in [26]). The accuracy of ASI given by [26] is 3.9%. Our uncertainty estimates for the merged SIC during spring are mostly at the upper end of the range of the passive microwave SIC algorithms given in [26], but higher in summer. However, the finer spatial resolution of our dataset has to be taken into account as an additional advantage. We thus judge our merged SIC uncertainty estimates as acceptable.

## 5. Summary

In this study, we first provide an assessment of the sensitivity of the MODIS SIC towards the choice of the ice tie-point, which is an essential parameter for the MODIS SIC retrieval. This is done by varying the starting position of the region which is used for the ice tie-point retrieval. This yields an ensemble of up to 48 possible MODIS ice tie-points, the mean of which is selected as final ice tie-point and the standard deviation of which is used as an estimate of the tie-point uncertainty. We find that the standard deviation averaged over all seasons is 0.33 K, which corresponds to 1.7% of the dynamic range. Converted to SIC, this yields an uncertainty of on average 1.9% and at maximum 6.2%. We mitigate this uncertainty as much as possible by varying the region for the ice tie-point retrieval and taking the average as ice tie-point.

Furthermore, our study presents an intercomparison of our merged SIC dataset and its constituents—the MODIS and AMSR2 SIC—with independent reference data. To this end, we produce a reference SIC dataset from 79 Sentinel-2 scenes by classifying the reflectances into water, thin ice and thick ice. The scenes cover the period from February 22nd to May 27th and are located in the Arctic marginal seas and in the Fram Strait (Figure 1). They are mostly dominated by a compact sea-ice cover and leads, but also comprise open-water areas. In theory, thin ice is part of the ice cover and should be reproduced as high SIC by a SIC retrieval. Thus, treating thin ice as ice (thin-ice SIC) is

the primary reference dataset. However, to evaluate the sensitivity of our SIC retrievals to thin ice, we created a second reference SIC dataset for which we included the thin ice in the open water class and only consider thick ice as ice (thick-ice SIC). Treating only thick ice as ice (thick-ice SIC) yielded a mean Sentinel-2 SIC of 87.3%, and treating thin and thick ice as ice (thin-ice SIC) yielded a mean SIC of 94.4%. In the latter case, the standard deviation decreases from 13.2% (thick-ice SIC) to 9.2%. The mean merged and MODIS SIC are 93.1% and 92.8%, respectively, which means that there is closer agreement with the thin-ice SIC than with the thick-ice SIC. Thus, both retrievals correctly identify thin ice as ice and only show a small underestimation of about 1% due to the presence of thin ice for the 66 Sentinel-2 reference scenes used for the intercomparison. The RMSD between the merged and MODIS SIC is 5%, which means that the algorithms do yield different results despite the small bias. We further investigate this by analysing one scene with good-quality MODIS SIC and one scene with poor-quality MODIS SIC. In the first case, the combination of the fine resolution of the MODIS SIC and the magnitude of the AMSR2 SIC used for the merged SIC retrieval allows better representation of the reference data than with either dataset alone. In this scene, the merged SIC has a smaller difference and a smaller RMSD than the MODIS SIC compared to the thin-ice reference dataset, while, at the same time, the fine resolution causes an open-water extent which is closer to the thin-ice SIC than the MODIS or AMSR2 SIC. In the poor-quality MODIS SIC case, an unscreened cloud and high ISTs deteriorate the quality of the MODIS SIC. This is mitigated by the merging, so that the result has a similar quality as the AMSR2 SIC alone, however likely not improving it.

In the third part of our study, we give an uncertainty estimate for the merged, MODIS and AMSR2 SIC. Uncertainties are derived by Gaussian error propagation based on the ice and water tie-point uncertainties and the observational uncertainty for the MODIS SIC and by expressing them as a function of the SIC according to [17] for the AMSR2 SIC. Gaussian error propagation then yields the merged SIC uncertainty. We use those uncertainties to determine a sanity threshold for the MODIS ice tie-point above which retrieval of meaningful MODIS SIC is no longer possible. A threshold of 266.5 K, i.e., about 5 K below the seawater freezing point, is chosen as a compromise between keeping the uncertainties small and discarding as few pixels as possible. A time-series of the uncertainties shows that the evolution of the merged and MODIS SIC uncertainty is mostly synchronous and between 5% and 15% until mid April, when the MODIS SIC uncertainty gets larger than the merged SIC uncertainty. In May, the average merged SIC uncertainty is 18% while the average MODIS SIC uncertainty is 25%. The AMSR2 SIC is constantly between 6% and 8% throughout the entire period, which is mostly due to our selection of scenes with high SIC and the much simpler uncertainty assumption for the AMSR2 SIC, which does not take atmospheric variability into account. Last, we compare our uncertainties to the differences between the merged, MODIS and AMSR2 SIC and the Sentinel-2 reference SIC (classifying thin ice as ice). We find that in most cases ($\approx$80%) our uncertainties are higher than the differences. We conclude that our uncertainties are a conservative estimate of the actual uncertainty.

## 6. Conclusions

We conclude from our study that the research questions raised in Section 1 can now be answered as follows.

1. How sensitive is the merged sea-ice concentration towards the choice of the MODIS ice tie-point? Choosing an arbitrary starting position for the ice temperature background field used as ice tie-point introduces a SIC uncertainty of 1.9%, which is mitigated by varying the starting position and taking the average of 48 iterations as ice tie-point. The standard deviation within these 48 iterations serves as uncertainty estimate for the MODIS ice tie-point.

2. How do the single-sensor and merged sea-ice concentration datasets compare with each other and the independently derived Sentinel-2 sea-ice concentration dataset? The mean merged and MODIS SIC are 93.1% and 92.8%, respectively, i.e., they agree within their uncertainties. The RMSD between the merged and the MODIS SIC amounts to 5%. They are closer to the reference SIC if thin ice is treated as ice (mean thin-ice SIC is 94.4%) than if it is

treated as water (mean thick-ice SIC is 87.3%). The slight negative bias for the merged and MODIS SIC compared to the thin-ice SIC but large positive bias compared to the thick-ice SIC means that most thin ice is correctly identified as ice and that thin ice is only causing a small, about −1% negative bias for the 66 cases which we evaluated. This, however, only applies as average for our cases, which on the other hand give a quite good spatial coverage of the winter time ice conditions along the Arctic marginal seas. For cases with extensive thin ice coverage the SIC underestimation can be larger.

3. What are the uncertainties of the single datasets?

   The MODIS and merged SIC uncertainties are close to each other and between 5% and 15% until the beginning of April, when the MODIS SIC uncertainty starts to increase more strongly. In May, the MODIS uncertainty is on average 25% and increases to 30%, while the mean merged SIC uncertainty is 18%. Thus, the merging significantly reduces the uncertainty later in the season. The AMSR2 SIC is mostly between 6% and 8%, but also uses a simpler uncertainty estimation. Comparison with the differences to the Sentinel-2 reference SIC shows that these differences are mostly smaller than the uncertainties, so that the uncertainties can be treated as a conservative estimate. In the majority of cases the accuracy of the merged SIC will be lower than the provided uncertainty estimate.

In summary, the benefit of combining MODIS and AMSR2 SIC becomes manifest in (a) a closer agreement with the reference SIC and lower uncertainty compared to the MODIS SIC, (b) a finer spatial resolution and enhanced potential to resolve leads compared to the AMSR2 SIC and (c) assuring spatial continuity in the presence of clouds, which is not possible with MODIS data alone.

## 7. Outlook

Evaluation in winter and in the Central Arctic could be done using SAR data. Retrieving SIC from SAR data is challenging and would require more analysis than we can provide in the framework of this paper. We recommend it as a direction for future research, especially as we expect a good performance of our merged dataset due to the large thermal contrast between water and sea ice in winter.

The meaningful use of MODIS thermal infrared data is limited to winter and spring conditions. In summer, visible data can provide similar fine-resolution data which can be used for the merging. Especially as most of the ship traffic is in summer, this would be a valuable extension of our dataset.

A study comparing our dataset to the coarser-resolution passive microwave SIC algorithms based on the 19 GHz and 37 GHz frequency channels could be done to demonstrate the advantage of the fine resolution of the merged SIC over these algorithms.

The MODIS instruments aboard Aqua and Terra have exceeded their design lifespans and successor missions like NASA's VIIRS (since 2011) aboard the Suomi NPP satellite or the SLSTR and OLCI on ESA's Sentinel-3A (since 2016) and Sentinel-3B (since 2018) satellites are already operational. Visible and thermal infrared data from these satellites can assure continuity of our merged dataset once MODIS ceases operation. A possible follow-up for the AMSR2 passive microwave data is the Copernicus Imaging Microwave Radiometer (CIMR) whose launch is scheduled for 2028.

**Author Contributions:** Conceptualization, V.L., G.S. and L.T.P.; Funding acquisition, G.S.; Investigation, V.L.; Methodology, V.L.; Resources, G.S.; Supervision, G.S. and L.T.P.; Validation, V.L.; Visualization, V.L.; Writing—original draft, V.L.; Writing—review & editing, G.S. and L.T.P. All authors have read and agreed to the published version of the manuscript.

**Funding:** This research was funded by the Institutional Strategy of the University of Bremen, funded by the German Excellence Initiative, and by the Deutsche Forschungsgemeinschaft (DFG) through the International Research Training Group IRTG 1904 ArcTrain and the Transregional Collaborative Research Center—TRR 172 'ArctiC Amplification: Climate Relevant Atmospheric and SurfaCe Processes, and Feedback Mechanisms', (AC)3 (project no. 268020496). The APC was funded by the University of Bremen.

**Acknowledgments:** We thank the editors and three anonymous reviewers for their constructive comments which greatly improved our manuscript. This research was funded by the Institutional Strategy of the University of Bremen, funded by the German Excellence Initiative, and by the Deutsche Forschungsgemeinschaft (DFG) through the International Research Training Group IRTG 1904 ArcTrain and the Transregional Collaborative Research Center—TRR 172 'ArctiC Amplification: Climate Relevant Atmospheric and SurfaCe Processes, and Feedback Mechanisms', (AC)3 (project no. 268020496). MODIS Ice Surface Temperature data were provided by the National Snow and Ice Data Center (NSIDC) at https://nsidc.org/data/MYD29/versions/6 (last access 12 September 2020). MODIS geolocation and cloud mask data were provided by the Level-1 and Atmosphere Archive & Distribution System (LAADS) Distributed Active Archive Center (DAAC) at https://ladsweb.modaps.eosdis.nasa.gov/archive/allData/61/MYD03/ (last access 12 September 2020). We thank JAXA (http://www.jaxa.jp/index_e.html) for the provision of AMSR2 data. We gratefully acknowledge the provision of Copernicus Sentinel data provided by the European Union via the Copernicus Open Access Hub (scihub.copernicus.eu).

**Conflicts of Interest:** The authors declare no conflict of interest.

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
