# Peer review of "Evaluation of a New Merged Sea-Ice Concentration Dataset at 1 km Resolution from Thermal Infrared and Passive Microwave Satellite Data in the Arctic"

_remotesensing, doi:10.3390/rs12193183_

Round 1
Reviewer 1 Report
A paper that proposes a method to enhance microwave SIC resolution with an aid of IR measurements. Results are apparently very encouraging.
I think that this approach is useful to determine the more precise and higher resolution of SIC information. But in the introduction, I cannot find why this research is important for the Arctic sea ice study.
It is well-known that IR measurements in Arctic winter are not likely to use for SIC retrieval because of the cloud mask. The authors made a short discussion regarding this, but more explanations and limitations of this study should be included. Additionally, the suggested method seems to be largely dependent on the accuracy level of the cloud mask. Quantitative error analysis due to uncertain cloud masks can be discussed in the manuscript.
From buoy measurements, you can easily check that water temperature over the Arctic Ocean is not considered to be constant, varying from -1.5 and -2.0oC depending on the salinity It will clearly affect the SIC product. Even if this assumption is not likely to affect the algorithm itself, the author should provide how this assumption is valid. A simple sensitivity test and/or error analysis is needed. Or you can provide proper references. The theoretical backgrounds are not well described in the manuscript and the variables are not well defined throughout the manuscript.
Specific Major comment:
- Page 6, “The AMSR2 SIC, on the other hand, are not as sensitive to thin ice and thus retrieve the right magnitude at a spatial scale of 5km and are available independently of cloud coverage.”
It seems to be not true. Most of the SIC algorithm is influenced by sea ice thickness variation when it is thin and young because thin ice is likely to be influenced by open water emission below the sea ice. Additionally, recent researches report that precipitating clouds make the SIC products more uncertain even in lower MW frequencies like C-band.
- Page 6 “ First, the AMSR2 data are resampled to a grid cell size of 1 km.”
How did you make downscaling data for AMSR2 measurements? I don't understand how you could resampling the AMSR2 data and then averaged over the original spatial scale. The author should provide a method to do that.
- Equations (5), (6), (7), and (8)
This equation system is not valid when Equations (6)-(8) are combined into Equation (5).
- Page 17, Lines 309-312.
Then, with this problem what is the advantage of these products when we take this product to use for other analyses?
Minor comment
- Page 2, Lines 48-50.
This sentence is hard to understand. Please rephrase it.
- Page 2, Lines 56-58.
The author can provide why thermal IR is more sensitive to sea ice thickness than passive microwave measurements.
- Page 3, Line 87
MODIS is already defined.
- Page 3, Line 120
Define ASI.
- Page 3, Line 127
Validation for what?
- Page 4, Table 1
Check again numbers in the table. It seems to be incorrect.
- Page 6, Equation 2
What is the ISTwater?
If it means open water surface temperature, the ISTwater can be just changed into SST or define it.
- Page 6, Line 147
What is the NSIDC projection, which is not a scientific.
- Page 6, “tend to underestimate the SIC due to the influence of thin ice thickness.”
Can you provide a one-sentence explanation of why thin ice thickness influences the SIC retrieval?
- Page 7, Equation (5)
Define Unc.
- Page 7, Equation (7)
SigmaSICMODIS may be changed into SigmaiceMODIS
- Figure 9
Same over the Kara Sea?
- Page 17, Line 335
Define ‘extent’.
- Table 2 & 3
‘xxx’ can be changed into “N/A”
Author Response
Dear Reviewer,
we are very grateful for your assessment of our paper. We have collected your comments and our responses in a PDF file, which we attach.
Best regards,
Valentin Ludwig (on behalf of the authors)

Reviewer 2 Report
The manuscript is overloaded with material including abstract and the number of figures.
It is necessary to shorten the text
Everything remove beginning "Appendix A" to "References".
Its don't have to theme of manuscript
Author Response
Dear Reviewer,
we are grateful for your assessment of our paper. We have collected your comments and our responses in a PDF file, which we attach.
Thanks a lot and best regards,
Valentin Ludwig (on behalf of the authors)

Reviewer 3 Report
I have enjoyed reading this manuscript and feel it presents an invaluable, merged sea ice concentration product that will greatly benefit the scientific community.
I deeply appreciate the effort to create a 1km Arctic sea ice concentration dataset using thermal infrared and passive microwave satellite data.
145: does the presence of liquid freshwater that may be at or slightly above 0C in melt ponds introduce much error here?
Figures 5 - 7 : look at improving the font size, reducing clutter on the axis if possible.
Line 395: I agree with keeping the leads as reduced SIC, but are they marked as 'leads' or merely just 'reduced SIC?"
Line 586: General question about the blended approach here: Is it possible to weight the two datasets in the merged product, co-varying with their uncertainty levels?
I don't see any discussion of meltponds in this manuscript, which would surely be appearing in areas by mid-late May. I recognize this gets us into the more uncertain time of year for passive microwave, but it would be good to have a bit of context regarding melt ponds & surface water discussed a bit more in the paper.
Author Response
Dear Reviewer,
we sincerely thank you for your assessment of our paper. We have collected your comments and our responses in a PDF file, which we attach.
Thanks a lot and best regards,
Valentin Ludwig (on behalf of the authors)

Round 2
Reviewer 1 Report
All my comments are well answered. The manuscript has gained a lot of improvement. finds the paper ready for publication.